# FRAMEORACLE: LEARNING WHAT TO SEE AND HOW MUCH TO SEE IN VIDEOS

## ABSTRACT

Vision-language models (VLMs) have advanced video understanding, but their performance is limited by the number of input frames they can process. Existing frame sampling strategies, such as uniform or fixed-budget selection, often fail to adapt to variations in information density or task complexity, resulting in inefficiency and information loss. To address this, we present **FrameOracle**, a lightweight and plug-and-play module that predicts both (1) which frames are most relevant to a given query and (2) how many frames are needed. FrameOracle is trained using a four-stage curriculum, with the first three stages relying on weak proxy signals such as cross-modal similarity. In the final stage, it leverages stronger supervision from a new dataset we introduce, **FrameOracle-41K**, the first large-scale VideoQA collection to provide keyframe annotations specifying the minimal set of frames required to answer each question. Extensive experiments across five VLMs and six benchmarks demonstrate that FrameOracle reduces 16-frame inputs to an average of 10.4 frames without any loss in accuracy. When starting from 64-frame candidates, it reduces the input to an average of 13.9 frames while improving accuracy by 1.4%, achieving state-of-the-art efficiency-accuracy trade-offs for scalable video understanding.

## 1 INTRODUCTION

Rapid advances in large language models (LLMs) (Stiennon et al., 2020; Gao et al., 2022; Yang et al., 2024) have enabled vision-language models (VLMs) to integrate visual understanding with strong linguistic reasoning (Zhang et al., 2024; Bai et al., 2025; Zhang et al., 2025a). This makes VLMs highly effective for complex video tasks such as question answering (Zhang et al., 2023; Lin et al., 2024a;b; Zhao et al., 2024; Xiao et al., 2025), summarization (Hua et al., 2025; Lee et al., 2025), and instruction following (Ren et al., 2024; Qian et al., 2024). A key challenge, however, is the large volume of data these models must process. Processing every video frame is computationally expensive, making efficient frame sampling essential (Hu et al., 2025). Most VLMs currently rely on simple approaches, such as uniform sampling at a fixed frame rate or selecting a fixed number of frames. While easy to implement, these methods have clear drawbacks: in long videos, they may miss crucial information, whereas in short videos, they often introduce redundant frames that waste resources, distract the model, and obscure key moments.

To mitigate this, a growing body of work has explored keyframe selection methods (Liu et al., 2025; Park et al., 2024; Tang et al., 2025; Zhang et al., 2025d). These approaches aim to identify a subset of frames that preserves semantic content while reducing redundancy. However, most existing methods assume a fixed, preset number of keyframes, ignoring the fact that the optimal number of frames varies across videos and queries. For example, short action-centric questions (e.g., whether a ball crosses a line in sports footage) may be resolved with just a handful of frames, while long-form narrative reasoning (e.g., inferring character intentions in a film) often requires a substantially larger set of frames. A few recent methods enable adaptive frame selection, but their adaptivity remains limited. In some cases, the selector is trained jointly with the backbone VLM (Buch et al., 2025), making it non-transferable to other models. In others, adaptivity is achieved via threshold-based filtering at inference, retaining only keyframes above a preset reward threshold. While this produces variable frame counts, it is not explicitly optimized during training, reducing effectiveness and generalizability. This raises a fundamental question: *How can we design a selector that identifies the*

*most relevant frames for a given query and determines how many are needed, while generalizing across different VLMs?*

To this end, we propose **FrameOracle**, a lightweight, plug-and-play frame selector that can be integrated with arbitrary VLMs. Unlike prior approaches that fix the number of frames in advance or require co-training with a specific backbone, FrameOracle jointly predicts (1) the importance of each frame relative to the query and (2) the number of frames to retain. The module is trained with a four-stage curriculum. The first three stages rely on weak proxy signals, such as cross-modal similarity and leave-one-out loss degradation. The final stage leverages stronger supervision from a new dataset we create, **FrameOracle-41K**, a large-scale VideoQA dataset with 40,992 examples and the first to provide keyframe annotations specifying the minimal frames required to answer each question. Unlike tasks such as object detection (Lin et al., 2014) or captioning (Xiong et al., 2024), no existing dataset provides ground-truth annotations identifying the keyframes. FrameOracle dynamically adapts its selections based on both video content and the prompt, operating seamlessly as a pre-processing module for any downstream VLM. In summary, our contributions are as follows:

- We propose **FrameOracle**, a lightweight and plug-and-play frame selector that dynamically predicts both which frames are most relevant and how many are needed.

- To facilitate training, we introduce **FrameOracle-41K**, the first large-scale VideoQA dataset with keyframe annotations, specifying the minimal set of frames needed to answer each question.

- We conduct extensive experiments across five VLMs and six benchmarks, showing that FrameOracle reduces 16-frame inputs to an average of 10.4 frames without any loss in accuracy. When starting from 64-frame candidates, it reduces the input to an average of 13.9 frames while improving accuracy by 1.4%, achieving state-of-the-art efficiency-accuracy trade-offs for video understanding.

## 2 RELATED WORK

**Keyframe Selection for Video Understanding.** Most existing keyframe selection methods assume a fixed frame budget: they rank candidate frames by visual–linguistic relevance or temporal salience and then retain the top-$k$ subset (Liang et al., 2024; Tan et al., 2024; Yu et al., 2025; Liu et al., 2025; Fang et al., 2025; Tang et al., 2025). Beyond this fixed-budget paradigm, some work has explored adaptive frame selection. These approaches fall into two categories. The first are agent-based methods, where large multimodal models act as decision-makers that iteratively analyze videos. For instance, VCA (Yang et al., 2025) combines curiosity-driven exploration with tree search to identify informative segments, while AKeyS (Fan et al., 2025) leverages a language agent to heuristically expand video segments and decide both which frames to retain and when to stop. However, such methods are computationally expensive due to repeated agent calls. The second category comprises approaches that require co-training with a specific VLM backbone (Buch et al., 2025; Yu et al., 2023; Guo et al., 2025), which restricts their portability. In contrast, FrameOracle is adaptive, lightweight, and model-agnostic: it learns to jointly predict which frames are relevant and how many to retain, while remaining plug-and-play across diverse VLMs.

**Datasets and Supervision for Video-Language Models.** Progress in video-language reasoning has been driven by large-scale datasets such as LLaVA-Video-178K (Zhang et al., 2024), ShareGPT4Video (Chen et al., 2024b), VideoRefer (Yuan et al., 2025), and CinePile (Rawal et al., 2024), which cover diverse scenarios and support both short- and long-form understanding. However, most of these datasets provide supervision only at the answer level, leaving the underlying evidence unannotated. In the absence of frame-level labels, keyframe selection methods are typically forced to rely on proxy signals, such as leave-one-out degradation or heuristic scoring. A few benchmarks, such as TVQA+ (Lei et al., 2020), ReXTime (Chen et al., 2024a), and HourVideo (Chandrasegaran et al., 2024), move toward span-level annotations, but none supply labels for both the indices of keyframes and the minimal sufficient number of frames needed to answer a question. FrameOracle-41K is the first dataset to provide explicit keyframe annotations for video–question pairs, offering high-quality supervision for both training and evaluation of adaptive frame selectors.

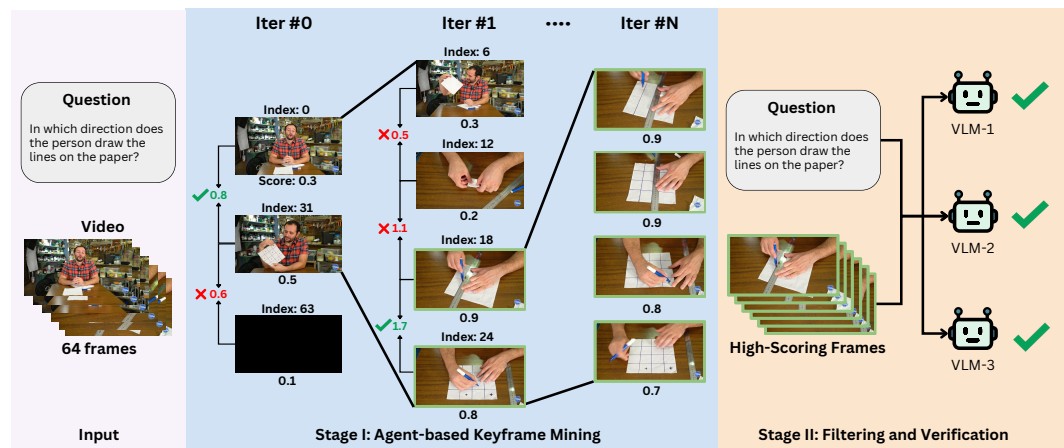

Figure 1: **FrameOracle-41K data generation pipeline.** Stage I (agent-based keyframe mining) iteratively explores each video using a multimodal agent, ultimately returning a predicted answer with confidence and relevance scores for all visited frames. Stage II (filtering and verification) first discards frames with low relevance scores and then verifies sufficiency by requiring three independent VLMs to answer correctly using only the remaining keyframes.

## 3 FRAMEORACLE-41K DATASET

We introduce FrameOracle-41K, the first VideoQA dataset that provides keyframe annotations, specifying the minimal set of frames needed to answer each question. The corpus contains 40,992 video–question pairs spanning diverse scenes and durations. In contrast to existing VideoQA datasets, which provide only ground-truth answers and, in some cases, coarse temporal spans in the video, FrameOracle-41K records, for each instance, the minimal number of frames needed to answer the question with high confidence, along with the keyframes that constitute the necessary evidence. Below, we describe our data generation pipeline, the verification and filtering procedures used to retain high-quality data, and the key statistics of the dataset.

### 3.1 DATA GATHERING AND PROCESSING

All video–question pairs in FrameOracle-41K are sourced from LLaVA-Video-178K (Zhang et al., 2024), a large-scale VideoQA dataset that covers a wide range of scenarios and activities. From this corpus, we first select nearly 100K videos, each 2–3 minutes long, balancing adequate temporal context with a manageable annotation effort. We then apply a two-stage process to create the final dataset (Figure 1). *Stage I (agent-based keyframe mining)* automatically extracts candidate keyframes using a multimodal agent that iteratively explores each video and assigns frame-level relevance scores. *Stage II (filtering and verification)* selects the minimal sufficient frame subset by retaining only samples where three independent VLMs consistently answer the question correctly. We further conduct a human verification on 4,000 randomly sampled instances, achieving an inter-annotator agreement of 94% and a verified accuracy of 93.3%. This confirms the reliability of the automatically generated annotations. Detailed procedures for each stage are described in the following paragraphs. Example JSON entries are in Appendix B, human verification protocol and results in Appendix E, and dataset visualizations in Appendix H.

**Stage I: Agent-based Keyframe Mining.** Starting from a uniformly sampled set of 64 frames, we employ an agent built on Qwen2.5-VL-72B API (Bai et al., 2025) to iteratively explore the video with respect to the given question. In the first iteration, the agent inspects three anchor frames (indices 0, 31, and 63), assigns relevance scores, and attempts an answer with a confidence estimate. It then compares the pairwise summed relevance of adjacent anchors (0+31 vs. 31+63) to decide which segment to explore next. Within the selected segment, a denser set of four anchor frames is sampled, another answer with confidence is attempted, and the same pairwise relevance comparison guides subsequent iterations. This iterative score–refine cycle continues until either the agent becomes confident enough to provide a stable answer or all frames have been examined. By the end of Stage I, the agent returns (1) its predicted answer and confidence, and (2) the complete set of

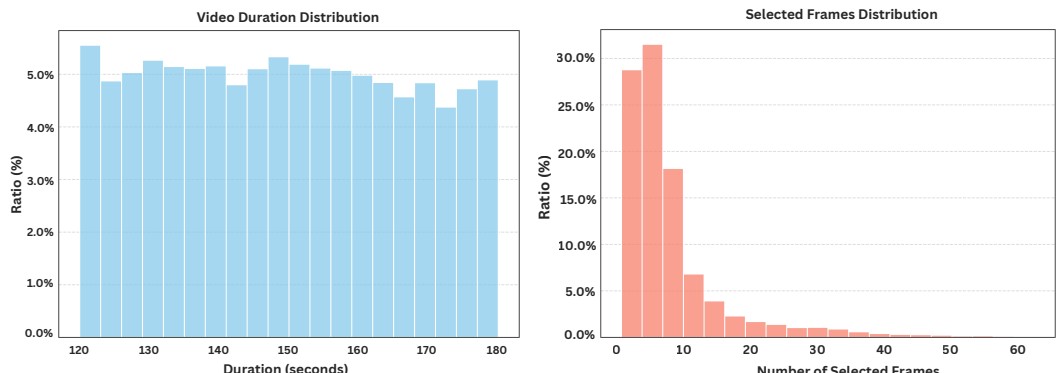

Figure 2: **FrameOracle-41K video-level statistics. Left:** Distribution of video durations. **Right:** Distribution of minimal sufficient keyframes per video–question pair.

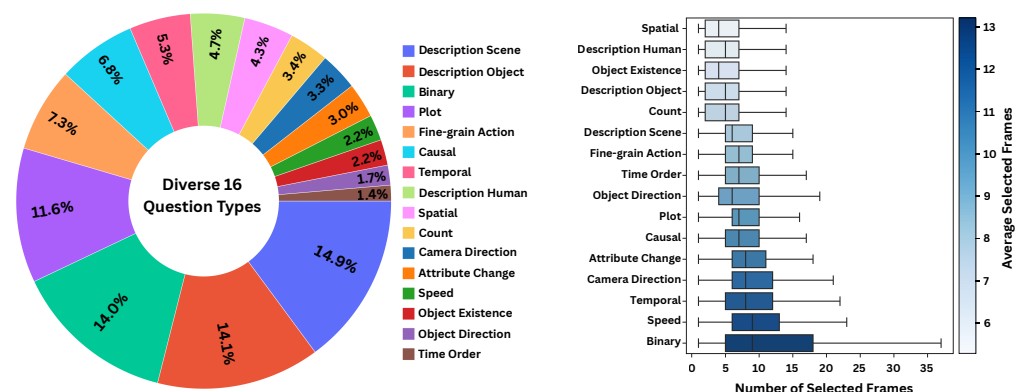

Figure 3: **FrameOracle-41K question-level statistics. Left:** Distribution of 16 question types across the dataset. **Right:** Per-type distributions of minimal sufficient keyframes.

frames it has inspected, each annotated with a relevance score. Any video-question pair for which the agent's predicted answer does not match the ground-truth answer is then discarded. This mined trajectory captures both the localization of question-specific evidence and fine-grained frame-level importance signals, forming the raw candidates for the next stage.

**Stage II: Filtering and Verification.** After obtaining candidate keyframes from Stage I, we first remove all frames with relevance scores below a threshold $\lambda$, leaving only those with stronger relevance. For each video–question pair, we then test whether the selected keyframes alone are sufficient to answer the question. Specifically, the keyframe set and the question are fed into three independent VLMs (i.e., Qwen2.5-VL-72B (Bai et al., 2025), LLaVA-OneVision-72B (Li et al., 2025), and LLaVA-Video-72B (Zhang et al., 2024)), and their predictions are compared against the ground-truth answer. Only instances for which all three models succeed using only the keyframes are retained. This cross-model verification ensures that the released dataset contains consistent, question-grounded keyframe annotations.

## 3.2 DATASET STATISTICS

Our two-stage pipeline produces 40,992 video–question pairs, forming the FrameOracle-41K dataset. Figure 2 (left) shows that most videos are two to three minutes long, providing sufficient temporal context without excessive redundancy. Figure 2 (right) shows the distribution of minimal sufficient keyframes per video–question pair: the median is five frames, the mean is around seven, and over 80% of samples require no more than 10 frames. A small fraction of more complex cases need 30 or more frames.

Figure 4: **Overview of the FrameOracle pipeline.** FrameOracle (dashed box) receives raw video frames and the textual prompt, and jointly predicts frame importance and the number of frames to keep. It outputs a compact keyframe subset, which is fed into the downstream VLM. $V_C$ denotes the pre-sampled frame collection, and $V_S$ denotes the subset selected by FrameOracle.

Figure 3 (left) categorizes all questions into 16 types following the taxonomy introduced in LLaVA-Video-178K (Zhang et al., 2024), covering a broad spectrum of reasoning skills such as description, localization, temporal understanding, and causal inference. Figure 3 (right) shows the per-type distribution of minimal keyframes. *Spatial* questions require the fewest frames (about 5.3 frames), while *Binary* questions need the most (around 13 frames), reflecting their underlying evidence needs. *Spatial* questions focus on static layouts within a scene, whereas *Binary* questions often ask whether an event occurs at any moment, requiring inspection of a broader temporal range. Some categories, such as *Camera Direction*, *Temporal*, *Speed*, and *Binary*, show high intra-class variability, with the number of required frames varying widely across instances. This indicates that even within a single reasoning type, temporal complexity and evidence density can differ significantly, highlighting the heterogeneous nature of FrameOracle-41K and motivating adaptive frame selection. Complete dataset statistics, question type definitions, and textual analyses are provided in Appendix D.

## 4 METHOD

We introduce **FrameOracle**, a lightweight and adaptive frame selector that dynamically determines the appropriate number of keyframes from a video, conditioned on the user prompt. FrameOracle enables efficient video understanding by providing the downstream VLM with a compact yet highly relevant subset of frames.

Since directly processing all frames of a video, $V$, is computationally expensive, we first apply uniform temporal sampling to extract a candidate set of $N$ frames, denoted as $V_C = \{f_1, \ldots, f_N\}$. This pre-sampling step acts as a coarse filter, reducing the input to a manageable size for FrameOracle (e.g., $N = 64$ or $N = 16$). Our goal is to learn FrameOracle, a *selection policy*, $\Pi_\theta$, parameterized by $\theta$, that operates on the candidate set. Given a candidate set $V_C$ and a text prompt $P$, FrameOracle selects a compact subset of frames $V_S \subset V_C$. Unlike approaches that fix the number of selected frames in advance, FrameOracle dynamically determines the subset size, $K = |V_S|$, as part of the selection process. Formally, $\Pi_\theta$ maps the pair $(V_C, P)$ to the selected subset $V_S$, which is then passed to a downstream VLM, $\mathcal{M}$, to perform a reasoning task, producing an output $A = \mathcal{M}(V_S, P)$. The objective is to train $\Pi_\theta$ to choose subsets that maximize the performance of $\mathcal{M}$ while keeping $K$ as small as possible.

### 4.1 FRAMEORACLE

FrameOracle, $\Pi_\theta$, is a neural module that learns to jointly predict frame importance and the number of frames to select from the candidate set $V_C$. We begin by extracting features from both the video frames and the text prompt, which serve as inputs to FrameOracle. For the candidate frame set $V_C$, we use a visual encoder to generate a sequence of $N$ frame embeddings. The text prompt $P$ is encoded using a tokenizer to obtain text embeddings. FrameOracle then operates on the projected embeddings and is fully agnostic to the underlying tokenizer. The FrameOracle architecture, shown in Figure 4, is composed of two main components: (1) a cross-modal fusion encoder and (2) dual prediction heads.

**(1) Cross-Modal Fusion.** To model the relationship between the text query and the video, we fuse the two modalities. Frame and text embeddings are first projected into a shared latent space using linear layers, and then processed by a stack of Transformer encoder layers. Specifically, we concatenate the projected text and frame embeddings, together with a learnable query token, into a single sequence $[\,k_{\text{query}};\,\text{text};\,\text{frames}\,]$, which is processed by the encoder whose self-attention performs token-level cross-modal interaction. Each frame is represented by one frame-level token, enabling efficient reasoning across text and frames with negligible computational cost relative to the downstream VLM.

**(2) Dual Prediction Heads.** The output of the fusion encoder is passed to two specialized heads, which form the core of our selection policy:

- **Rank Head:** This head evaluates the relevance of each candidate frame to the prompt. It processes the fused feature sequence to output a scalar importance score, $s_i$, for each frame $f_i \in V_C$, resulting in a score vector $S = \{s_1, \ldots, s_N\}$.

- **K Head:** This head predicts how many frames to select from the candidate set. It takes the globally aggregated features from the fusion encoder and outputs a probability distribution over a discrete set of possible values for $K$, where $K \leq N$.

## 4.2 TRAINING

We train FrameOracle using a curriculum-based, four-stage protocol. This strategy progressively refines the policy $\Pi_\theta$, teaching it to reason effectively over the pre-sampled frames (e.g., 16 or 64). The staged training leverages four widely used public VideoQA datasets, covering clips ranging from roughly 10 seconds to 15 minutes in length. Details of the full dataset composition are provided in Appendix A.

**Stage 1: Text-Visual Alignment.** The initial stage focuses on learning a robust cross-modal representation by aligning the textual prompt with the visual content of the candidate frames. We use the pre-trained text-visual model SigLIP (Zhai et al., 2023) as a teacher to provide supervision. For each prompt-frame pair, a SigLIP similarity score serves as the target relevance signal. The feature projectors and the cross-modal Transformer encoder are trained with a RankNet loss (Burges et al., 2005), encouraging the model's predicted scores to match the relative ordering of the SigLIP similarities. Concretely, for any two frames $i$ and $j$, let $s_i$ and $s_j$ denote their SigLIP similarity scores, and $y_i$ and $y_j$ their predicted scores. We define the pairwise preference label as $t_{ij} = \text{sign}(s_i - s_j)$. The RankNet loss is then given by

$$\mathcal{L}_{\text{RankNet}} = \sum_{i<j} \log\Big(1 + \exp\big(-t_{ij}\,(y_i - y_j)\big)\Big), \tag{1}$$

where $t_{ij} = 0$ corresponds to tied frames and does not contribute to the gradient. In this way, the alignment capability of SigLIP is distilled into FrameOracle. The K Head remains frozen during this stage.

**Stage 2: Rank Head Optimization.** In the second stage, we train the Rank Head to identify the most salient frames in the candidate set $V_C$. Unlike the first stage, where SigLIP-based supervision is computed independently for each frame and provides no temporal guidance, this stage uses the downstream VLM's loss as a supervisory signal, allowing the selector to capture temporal dependencies across frames. To generate training targets, we adopt a leave-one-out (LOO) approach: for each frame $f_i \in V_C$, we remove it from the input set and pass the remaining frames through the VLM, measuring the change in its loss. A larger increase indicates that $f_i$ is more important. These importance scores serve as soft targets, and the Rank Head is trained with a RankNet loss to predict them. During this stage, the K Head remains frozen, while the Transformer encoder and feature projectors are fine-tuned with a smaller learning rate to stabilize training.

**Stage 3: K Head Optimization.** The third stage focuses on training the K Head to predict the number of frames. During this stage, the Rank Head is frozen, while the feature projectors and the Transformer encoder are fine-tuned with a very small learning rate for slight adaptation. For each sample, we evaluate the downstream VLM (the same backbone and task loss as in Stage 2) using the top-$k$ frames from $V_C$, ranked by the frozen Rank Head, for a candidate value $k \in N$. We then

select the target

$$k^* = \arg\min_{k \in N} \Big( \mathrm{zscore}(\mathcal{L}_{\mathrm{task}}(k)) + \lambda_k \, k \Big), \tag{2}$$

where the linear penalty balances accuracy and frame cost. The K Head predicts a categorical distribution $p_\theta(k)$ over $k \in \{1, \ldots, N\}$ and is trained with

$$\mathcal{L}_K = (1 - \alpha)\,\mathcal{L}_{\mathrm{evo}} + \alpha\,\mathcal{L}_{\mathrm{class}}, \tag{3}$$

where the Expected Value Objective

$$\mathcal{L}_{\mathrm{evo}} = \mathrm{SmoothL1}\left( \sum_{k=1}^{N} k\, p_\theta(k),\ k^* \right)$$

regresses the predicted expectation to $k^*$, and $\mathcal{L}_{\mathrm{class}}$ is a KL divergence aligning $p_\theta$ with a Gaussian-shaped soft target centered at $k^*$.

**Stage 4: Supervised Fine-tuning with Ground Truth.** In the final stage, we perform supervised fine-tuning (SFT) on FrameOracle-41K, which provides supervision for both the keyframe indices and the number of frames. Unlike the earlier stages, which rely on weak or proxy signals, FrameOracle-41K offers high-quality annotations that have been verified for consistency. The Rank Head is trained to align its predictions with the annotated keyframes, while the K Head is jointly trained to match the annotated $K$ values. This strong, direct supervision further refines the selection policy beyond what is achieved in Stages 1–3.

## 5 EXPERIMENTS

### 5.1 EXPERIMENT SETTINGS

**Implementation Details.** We train two versions of FrameOracle, using uniformly sampled frames as selector inputs: one with 16 frames and another with 64 frames, respectively. Both of them use DINOv2 (Oquab et al., 2024) as the visual encoder and Qwen2.5-VL as the tokenizer. Training follows the four-stage curriculum described in Section 4.2, progressively optimizing the Rank Head and K Head using proxy signals and FrameOracle-41K annotations. In Stages 2 and 3, we adopt Qwen2.5-VL-3B as the backbone VLM for supervision. For the 64-frame selector, we additionally cap the maximum predicted $K$ at 16 during Stage 3 to ensure comparability with the experimental settings. All experiments are run on 8×H100 GPUs. Detailed hyperparameter settings, including learning rates, batch sizes, and training durations for each stage, are provided in Appendix A.

**Benchmarks.** We evaluate FrameOracle on six widely adopted video benchmarks, which can be divided into long-video and short-video understanding tasks. For long-video understanding, we include EgoSchema (Mangalam et al., 2023), LongVideoBench (Wu et al., 2024), MLVU (Zhou et al., 2025), and Video-MME (Fu et al., 2025), all of which require reasoning over extended temporal contexts ranging from minutes to hours. These datasets emphasize challenges such as cross-event reasoning, global consistency, and temporal grounding across lengthy sequences. For short-video understanding, we evaluate on NExTQA (Xiao et al., 2021), and Perception (Pătrăucean et al., 2023), which involve clips typically within tens of seconds. These benchmarks focus on fine-grained event recognition, local temporal relations, and reasoning within concise videos. We follow the LMMs-Eval library (Zhang et al., 2025b) for evaluation, and report accuracy across all benchmarks.

### 5.2 RESULTS AND ANALYSIS

**Comparisons with State-of-the-Art Models.** Table 1 presents a comprehensive comparison of FrameOracle across two categories: (1) five state-of-the-art (SOTA) VLMs, and (2) its integration with six diverse VLMs, Qwen2.5-VL (Bai et al., 2025), LLaVA-OneVision (Li et al., 2025), LLaVA-Video (Zhang et al., 2024), VideoLLaMA3 (Zhang et al., 2025a), Qwen3-VL(QwenTeam, 2025), and the proprietary GPT-4o (Hurst et al., 2024). Qwen-VL series internally merges every two adjacent frames into a single representation. To ensure a fair comparison with models that process raw frames directly, we report the baseline using 32 frames. For each model integrated with FrameOracle, we report results for two configurations: using 16-frame FrameOracle and 64-frame FrameOracle.

Table 1: **FrameOracle vs. SOTA VLMs.** "Frames" shows $M \to \bar{K}$: FrameOracle starts from $M$ uniformly sampled frames and reduces to an average of $\bar{K}$. Highlighted rows show the upper-bound performance with larger frame inputs. LVB = LongVideoBench validation set.

| Model | Frames | NExTQA | | | Perception | LVB | Video-MME | EgoSchema | MLVU | Avg. |
|---|---|---|---|---|---|---|---|---|---|---|
| | | OE_val | OE_test | MC | | | | | | |
| *(1) State-of-the-Art Models* | | | | | | | | | | |
| ShareGPT4Video-8B (Chen et al., 2024b) | 16 | - | - | - | - | 41.8 | 39.9 | - | 46.4 | - |
| LLaMA-VID-7B (Li et al., 2024b) | 16 | - | - | - | 44.6 | - | 25.9 | 38.5 | 33.2 | - |
| VideoChat2-7B (Li et al., 2024a) | 16 | - | - | - | - | 39.3 | 39.5 | 63.6 | 44.5 | - |
| VideoLLaMA2-7B (Cheng et al., 2024) | 16 | - | - | 45.4 | 54.9 | 53.1 | 47.9 | 53.1 | - | - |
| InternVL2-40B (OpenGVLab, 2024) | 16 | - | - | - | - | 59.3 | 61.2 | - | 59.5 | - |
| *(2) FrameOracle on Different Baselines* | | | | | | | | | | |
| Qwen2.5-VL-3B (Bai et al., 2025) | 32 | 25.1 | 29.6 | 75.4 | 65.9 | 54.1 | 58.4 | 53.4 | 59.4 | 52.7 |
| + FrameOracle | 32→20.9 | 25.6 | 30.5 | 74.8 | 66.7 | 54.3 | 58.5 | 53.8 | 58.4 | 52.8 |
| + FrameOracle | 128→27.8 | **26.0** | **31.7** | **76.1** | **67.8** | **54.8** | **59.7** | **54.5** | **61.6** | **54.0** |
| LLaVA-OneVision-7B (Li et al., 2025) | 16 | 14.6 | 16.7 | 78.2 | 56.4 | 55.0 | 56.1 | 60.8 | 60.9 | 49.8 |
| + FrameOracle | 16→10.4 | 16.1 | 17.8 | 77.6 | 56.5 | 55.5 | 56.0 | 62.4 | 60.2 | 50.3 |
| + FrameOracle | 64→13.9 | **16.5** | **19.0** | **78.5** | **56.9** | **56.5** | **58.1** | **63.4** | **63.7** | **51.6** |
| LLaVA-Video-7B (Zhang et al., 2024) | 16 | 27.3 | 32.4 | 81.0 | 64.3 | 55.8 | 59.8 | 54.2 | 61.7 | 54.6 |
| + FrameOracle | 16→10.4 | 27.8 | 33.0 | 80.4 | 64.7 | 56.3 | 59.6 | 54.6 | 60.8 | 54.7 |
| + FrameOracle | 64→13.9 | **28.8** | **33.9** | **81.6** | **65.1** | **57.8** | **61.6** | **55.2** | **64.3** | **56.0** |
| VideoLLaMA3-7B (Zhang et al., 2025a) | 16 | 27.8 | 32.3 | **82.3** | 72.3 | 56.1 | 61.2 | 61.4 | 50.9 | 55.5 |
| + FrameOracle | 16→10.4 | 28.3 | 32.9 | 81.2 | 72.0 | 56.0 | 61.4 | 61.8 | 52.8 | 55.8 |
| + FrameOracle | 64→13.9 | **28.9** | **33.6** | 82.0 | **72.8** | **56.9** | **61.8** | **62.4** | **54.1** | **56.6** |
| Qwen3-VL-8B (QwenTeam, 2025) | 32 | 26.0 | 31.1 | 76.6 | 67.5 | 63.3 | 66.9 | 70.8 | 63.6 | 58.2 |
| + FrameOracle | 32→20.9 | 26.6 | 32.3 | 76.1 | 68.2 | 64.0 | 67.3 | 71.4 | 62.9 | 58.6 |
| + FrameOracle | 128→27.8 | **28.1** | **33.8** | **77.3** | **69.0** | **65.2** | **69.1** | **72.3** | **66.3** | **60.1** |
| GPT-4o (Hurst et al., 2024) | 16 | - | - | **63.1** | - | 51.6 | 58.5 | 66.0 | **38.7** | 55.6 |
| + FrameOracle | 16→11.1 | - | - | 62.9 | - | **52.1** | **59.2** | **68.8** | 38.1 | **56.2** |

Under the 16-frame condition, FrameOracle maintains accuracy comparable to the baseline models across all benchmarks while reducing the number of frames by approximately 35%. With 64-frame inputs, FrameOracle begins with a denser candidate set and adaptively selects relevant frames. In this setting, it consistently improves performance over the baseline models while still reducing frames by about 15%. This demonstrates that a larger candidate pool enables FrameOracle to better exploit temporal redundancy, resulting in improved accuracy–efficiency trade-offs. FrameOracle is trained independently and applied in a fully plug-and-play manner, requiring no co-training or backbone-specific adaptation. These results confirm its ability to generalize across model architectures.

**RQ 1: Does giving a VLM more frames consistently improve its performance?**

One might expect that providing more frames always improves performance, since additional frames offer more visual evidence. However, Table 1 shows the opposite: using more frames often fails to help and can even reduce accuracy. This aligns with recent findings that long-video reasoning is inherently sparse, with only a small subset of frames being truly relevant (Park et al., 2024). Extra frames primarily introduce redundancy and noise, diluting cross-modal attention and yielding diminishing returns (Li et al., 2023).

By contrast, when FrameOracle selects a smaller but more informative subset of frames, performance can improve, especially on open-ended benchmarks. For example, on LLaVA-OneVision-7B, reducing 16 frames to roughly 10.4 improves NExTQA metrics (OE_val: $14.6 \to 16.1$, OE_test: $16.7 \to 17.8$) and EgoSchema ($60.8 \to 62.4$). Similar trends are observed for GPT-4o, where accuracy rises from 55.6 to 56.2 despite using fewer frames. Qualitative examples in Appendix I (Figure 9) further illustrate this effect: FrameOracle identifies the key evidence with fewer frames, yielding correct answers where naive higher-frame sampling fails.

Crucially, this improvement does not come from simply reducing the visual input size. As shown in our ablation (Appendix F.3), uniformly sampling 10 frames from the same 16-frame input reduces performance substantially ($49.8\% \to 46.3\%$ on LLaVA-OneVision). In contrast, FrameOracle improves accuracy to 50.3% with the same 10-frame budget. This demonstrates that the gain comes from selecting semantically relevant frames, not from alleviating token overload.

**RQ 2: How does FrameOracle compare with existing SOTA methods for keyframe selection?**

We compare FrameOracle with (1) keyframe selection methods that are jointly trained with their VLM backbone in their original configurations, and (2) plug-and-play keyframe selection methods applied to open-source models (Table 2). The first category cannot be applied directly to open-source models, and FFS (Buch et al., 2025) is the only method that adaptively determines the number of

Table 2: **FrameOracle vs. SOTA keyframe selection methods.** NExTQA reports MCQ. Methods using more frames or larger LLMs are shown in gray. LVB = LongVideoBench validation set.

| Model | Frames | NExTQA | LVB | Video-MME | EgoSchema | MLVU |
|---|---|---|---|---|---|---|
| *(1) Jointly Trained Keyframe Selection Methods* | | | | | | |
| SeViLA (Yu et al., 2023) | 8 | 63.6 | - | - | 25.7 | - |
| LVNet (Park et al., 2024) | 12 | 72.9 | - | - | - | - |
| VideoAgent (Wang et al., 2024) | 8.4 | 71.3 | - | - | 60.2 | - |
| FFS (Buch et al., 2025) | 8.6 | 66.7 | - | - | - | - |
| MoReVQA (Min et al., 2025) | 30 | 69.2 | - | - | - | - |
| VSLS (Guo et al., 2025) | 32 | - | 63.4 | 63.0 | - | - |
| AKS (Tang et al., 2025) | 64 | - | 62.7 | 65.3 | - | - |
| *(2) Plug-and-Play Keyframe Selection Methods* | | | | | | |
| LLaVA-OneVision-7B (Li et al., 2025) | 8 | 77.4 | 54.3 | 53.8 | 62.0 | 58.4 |
| + Frame-Voyager (Yu et al., 2025) | 128→8 | 73.9 | - | **57.5** | - | **65.6** |
| + BOLT (Liu et al., 2025) | 1fps→8 | 77.4 | 55.6 | 56.1 | 62.2 | 63.4 |
| + KFC (Fang et al., 2025) | 1fps→8 | - | 55.6 | 55.4 | - | 65.0 |
| **+ FrameOracle** | 64→8 | **77.8** | **56.0** | **57.5** | **62.8** | 62.9 |
| LLaVA-Video-7B (Zhang et al., 2024) | 8 | 75.6 | 54.2 | 55.9 | 51.8 | 60.5 |
| + BOLT (Liu et al., 2025) | 1fps→8 | - | - | 58.6 | - | - |
| + KFC (Fang et al., 2025) | 1fps→8 | - | 56.5 | 57.6 | - | **66.9** |
| **+ FrameOracle** | 64→8 | **76.5** | **56.9** | **58.9** | **53.0** | 63.4 |

Table 3: **Comparison of FLOPs, latency, visual tokens, and accuracy.** The values of the computational cost are reported as per-GPU, per-sample averages.

| Model | Frames | TFLOPs ↓ | | | | Latency (s) ↓ | Visual Tokens ↓ | Avg. Acc. ↑ |
|---|---|---|---|---|---|---|---|---|
| | | DINOv2 | FrameOracle | VLM | Total | | | |
| LLaVA-Video-7B | 16 | – | – | 184.38 | 184.38 | 0.615 | 11,644.0 | 54.6 |
| + FrameOracle | 16→10.4 | 1.87 | $2.6 \times 10^{-4}$ | **109.11** | **110.98** | **0.363** | **7,581.6** | 54.7 |
| + FrameOracle | 64→13.9 | 7.58 | $1.0 \times 10^{-3}$ | 160.09 | 167.67 | 0.556 | 10,133.1 | **56.0** |

retained frames; all other methods assume a fixed number of keyframes. All plug-and-play selection methods only provided 8-frame results. To ensure a fair comparison, we disable FrameOracle's K Head and rely solely on the Rank Head: given 64 uniformly sampled frames, we select the top-8 ranked frames as input to the backbone VLMs.

FrameOracle achieves competitive performance compared to prior plug-and-play methods. Across NExTQA, LongVideoBench, Video-MME, and EgoSchema, it improves accuracy by roughly 2–4 percentage points, outperforming Frame-Voyager (Yu et al., 2025), BOLT (Liu et al., 2025), and KFC (Fang et al., 2025). On MLVU, FrameOracle outperforms the base VLMs but does not surpass heuristic methods such as KFC, a greedy selection strategy that maximizes relevance and diversity, which achieve higher scores. This gap reflects MLVU's focus on fine-grained temporal grounding and multi-event reasoning, where heuristics can sometimes capture domain-specific cues more effectively. Overall, the results demonstrate that even without the K Head, the Rank Head alone can reliably prioritize important frames and deliver consistent gains across multiple VLMs, achieving state-of-the-art performance on most benchmarks. Beyond plug-and-play selectors, we also compare FrameOracle with memory-based video compression methods such as MovieChat (Song et al., 2024) on long-video benchmarks. As shown in Appendix F.6, when evaluated under the same downstream VLM (LLaVA-OneVision) and the same inference setup, FrameOracle achieves comparable or better performance while using significantly fewer frames, highlighting its efficiency in handling long temporal contexts.

**RQ 3: How much can FrameOracle reduce computational cost while preserving accuracy?**

We take LLaVA-Video-7B with 16 input frames as the baseline and report per-GPU, per-sample averages. FrameOracle reduces the input from 16 to 10.4 frames, cutting the VLM cost from 184.38 to 109.11 TFLOPs and the end-to-end total from 184.38 to 110.98 TFLOPs ($-39.8\%$). It also lowers latency from 0.615 to 0.363 seconds ($-41.0\%$) and reduces tokens from 11,644.0 to 7,581.6, while maintaining accuracy. With a larger candidate pool, FrameOracle reduces 64 frames to 13.9, improving accuracy by $+1.4$ while still lowering total compute to 167.67 TFLOPs ($-9.1\%$), tokens

Table 4: **Four-stage training of FrameOracle, evaluated on Qwen2.5-VL-3B.** Stages are added progressively to assess their impact. The baseline (first row) randomly selects 16 of 32 frames. **Bold** numbers indicate best performance. LVB = LongVideoBench validation set.

| Model | Frames | NExTQA | | | Perception | LVB | Video-MME | EgoSchema | MLVU |
|---|---|---|---|---|---|---|---|---|---|
| | | OE_val | OE_test | MC | | | | | |
| Qwen2.5-VL-3B | 32→16 | 23.4 | 29.1 | 71.9 | 65.0 | 52.9 | 54.8 | 50.2 | 56.7 |
| + Stage 1 | 32→16 | 24.7 | 29.2 | 72.4 | 60.3 | 49.8 | 52.4 | 48.2 | 51.3 |
| + Stage 2 | 32→16 | 24.8 | 29.5 | 73.0 | 64.7 | 51.9 | 55.7 | 52.2 | 54.8 |
| + Stage 3 | 32→21.8 | 25.1 | 30.0 | 74.1 | 66.0 | 53.7 | **59.4** | 53.6 | 57.6 |
| + Stage 4 | 32→20.9 | **25.6** | **30.5** | **74.8** | **66.7** | **54.3** | 58.5 | **53.8** | **58.4** |

to 10,133.1 ($-13.0\%$), and latency to 0.556 seconds ($-9.6\%$). These results reveal a clear trade-off: smaller frame pools yield larger efficiency gains without harming accuracy, while larger pools provide accuracy improvements with moderate compute savings. Although this may appear to show diminishing efficiency returns, roughly 90% of the total computation comes from the backbone VLM. As shown in Table 3, moving from the $16 \to 10.4$ to the $64 \to 13.9$ setting increases total FLOPs almost entirely due to the VLM, while the selector accounts for only about 10%. Similar efficiency–accuracy trade-offs should hold for other $\sim$7B-scale models, with only minor variations across architectures.

**RQ 4: Are all training-stage components essential for FrameOracle's performance?**

We conduct ablations over the four training stages to evaluate the contribution of each stage, using Qwen2.5-VL-3B as the backbone VLM (Table 4). The baseline (first row) randomly selects 16 frames from 32 uniformly sampled candidates. Stage 1 (text–visual alignment) underperforms the baseline, though it provides a foundational starting point. Stages 2 (Rank Head) and 3 (K Head) yield clear performance improvements, and the full model with Stage 4 (fine-tuning on FrameOracle-41K) delivers further gains over Stages 2 and 3 on most benchmarks, with only a slight decline on Video-MME. Moreover, the average number of retained frames decreases from 21.8 to 20.9, showing that FrameOracle-41K supervision stabilizes performance while enabling higher accuracy with fewer frames. These results demonstrate that ground-truth supervision from FrameOracle-41K is essential for refining both frame importance scoring and the prediction of the number of frames, establishing it as a valuable resource for adaptive frame selection. Furthermore, we validate the necessity of the four-stage curriculum via two ablation studies. As detailed in Appendix F.1, we find that skipping intermediate stages (e.g., applying Stage 4 directly after Stage 1) leads to overfitting and reduced performance, while jointly optimizing Stage 2 and 3 results in training collapse and performance drop due to mutual interference between the ranking and budgeting heads. These results confirm that the progressive design is essential for both stable optimization and robust generalization.

# 6 CONCLUSION

We propose FrameOracle, a lightweight, plug-and-play frame selector that adaptively determines which frames to retain and how many are needed. To facilitate training, we introduce FrameOracle-41K, a large-scale VideoQA dataset with 40,992 examples, and the first to provide keyframe annotations specifying the minimal frames required to answer each question. Experiments show that FrameOracle improves diverse VLM backbones without co-training, reducing FLOPs, latency, and token usage, while outperforming state-of-the-art keyframe selection methods. Future work will explore supporting variable-sized frame inputs.

REPRODUCIBILITY STATEMENT

To support reproducibility, we provide details on both the model and dataset. FrameOracle's design, including its learning objective, selection policy, and staged curriculum, is described in Section 4, with training procedures and hyperparameters in Appendix A (Figure 5; Table 5). FrameOracle-41K construction, including agent-based mining, verification, and dataset format, is covered in Section 3 and Appendix B. Evaluation settings, including benchmarks, backbones, and metrics, are in Section 5.1. These sections provide all the information needed to reproduce our results.

ETHICS STATEMENT

FrameOracle-41K is created from source videos collected from the internet, which may contain images of individuals and reflect societal biases present in online content. Our data processing pipeline does not involve identifying or profiling any individuals. The data is used solely for developing our video understanding model. We release the dataset strictly for non-commercial, academic research purposes and caution future users to be aware of potential inherent biases in the data.

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

# APPENDIX

## A  FULL IMPLEMENTATION DETAILS

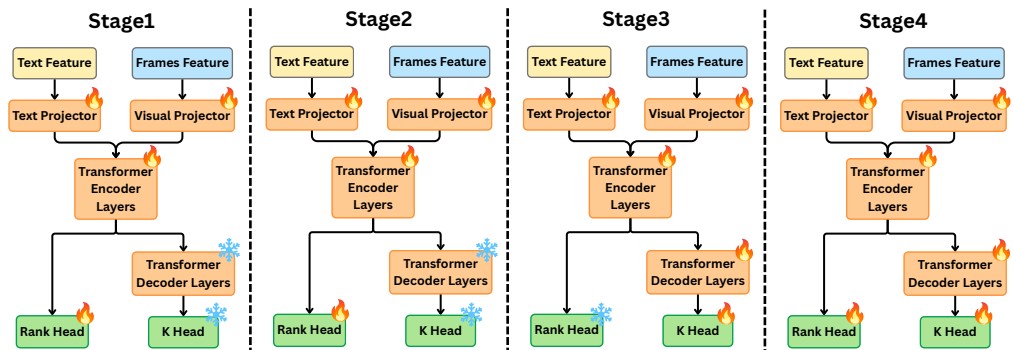

Figure 5: **Four-stage training strategy of FrameOracle.** The model is progressively optimized from weak to strong supervision, culminating in supervised fine-tuning with FrameOracle-41K annotations. Fire icons indicate trainable modules, while snowflake icons denote frozen ones.

Table 5: **Datasets used in FrameOracle training.** Stage 4 leverages FrameOracle-41K.

| Task | Dataset | Amount |
|------|---------|--------|
| Stage1 | LLaVA-Video-178K (Zhang et al., 2024), ShareGPT4o-Video (Chen et al., 2024b), Video-ChatGPT (Maaz et al., 2024) | 300K |
| Stage2 | LLaVA-Video-178K (Zhang et al., 2024), LLaVA-Hound (Zhang et al., 2025c), Video-ChatGPT (Maaz et al., 2024) | 300K |
| Stage3 | LLaVA-Video-178K (Zhang et al., 2024), LLaVA-Hound (Zhang et al., 2025c), Video-ChatGPT (Maaz et al., 2024) | 300K |
| Stage4 | **FrameOracle-41K (Our Dataset)** | 40K |

**Training strategy illustration.** Figure 5 presents a schematic of our four-stage curriculum, highlighting trainable modules (fire) and frozen modules (snowflake) at each stage.

**Hardware and input budgets.** All training is conducted on 8×H100 GPUs. We train two FrameOracle variants: one with 16 uniformly sampled candidate frames and another with 64. A cosine learning rate scheduler with the AdamW optimizer is used across all stages.

**Datasets used in staged training.** FrameOracle is optimized using a four-stage curriculum with progressively stronger supervision. Stages 1-3 rely on large-scale video–language corpora, while Stage 4 leverages our FrameOracle-41K dataset. Table 5 summarizes the dataset composition for each stage.

**Stage 1: Cross-modal alignment.** K Head is frozen while the feature projectors and cross-modal Transformer encoder are trained jointly, both optimized with a learning rate of $1 \times 10^{-4}$. The 16-frame selector uses a batch size of 16 and trains for approximately 48 hours, whereas the 64-frame version uses a smaller batch size of 2 and completes in about 91 hours.

**Stage 2: Rank Head optimization.** Rank Head is trained while the K Head remains frozen. The Rank Head uses a learning rate of $1 \times 10^{-4}$, and the feature projectors and Transformer encoder are fine-tuned with a smaller learning rate of $1 \times 10^{-5}$. The 16-frame selector uses a batch size of 16 and trains for approximately 40 hours, whereas the 64-frame variant uses the same batch size and takes about 52 hours.

**Stage 3: K Head optimization.** K Head is the primary trainable module, optimized with a learning rate of $1 \times 10^{-4}$. The feature projectors and Transformer encoder are lightly updated with a learning rate of $1 \times 10^{-7}$, while the Rank Head remains frozen. We set $\lambda_k = 0.0105$ to balance accuracy

and efficiency. The 16-frame selector uses a batch size of 16 and trains for approximately 35 hours, whereas the 64-frame variant uses the same batch size and takes about 60 hours.

**Stage 4: Supervised fine-tuning on FrameOracle-41K.** Rank Head and K Head are trained jointly with a learning rate of $5 \times 10^{-5}$, while the feature projectors and Transformer encoder are fine-tuned with $1 \times 10^{-5}$. The 16-frame selector is trained with a batch size of 8 for approximately 12 hours, and the 64-frame version uses the same batch size and trains for about 18 hours.

## B  FRAMEORACLE-41K DATA FORMAT

We release the FrameOracle-41K dataset in JSON format, with each entry corresponding to a single video–question pair. Each entry includes the instance identifier, question–answer pair, paths to the associated video and extracted keyframes, video duration, and number of selected frames. Below, we provide an example JSON entry to illustrate the dataset's structure.

```
{
  "id": 30,
  "question": "What folding technique is demonstrated first in the video?
      ",
  "ground_truth_answer": "The 'SHIKAKU NO GI' (Square Fold) technique is
      demonstrated first.",
  "video": "/srv/nfs/video_data/video/ytb_8yhoV5C3bT8.mp4",
  "keyframes_dir": "/srv/nfs/video_data/extracted_frames/ytb_8yhoV5C3bT8"
      ,
  "duration": 126.893,
  "num_selected_frames": 8
}
```

## C  PROMPTS FOR DATA GENERATION

---

**Prompt Template for Stage I: Initial Frame Analysis**

You are analyzing a video that is {duration_seconds} seconds long. The video has been uniformly sampled into 64 frames, indexed from 0 (start) to 63 (end).

Analyze these {len(initial_indices)} initial frames (indices: {initial_indices}) to answer: "{question}". Provide a short caption for each frame, a relevance score (INTEGER 1-5), your confidence (high/medium/low), and your answer attempt.

Respond in JSON: {{"frame_analysis": [{{"index": int, "caption": "str", "relevance": int}}], "confidence": "str", "answer_attempt": "str", "reasoning": "str"}}

**IMPORTANT GUIDELINES:**

- Relevance combines BOTH
(a) how well the frame's TEMPORAL POSITION matches the question mentioned, and
(b) how much the visible CONTENT answers the question. A high score (4-5) requires strong evidence on both axes.
- You may use "high" confidence early ONLY IF: You have seen explicit, definitive evidence that unquestionably answers the question (e.g., clearly visible target object/person/action).
- Before setting "high" confidence, explicitly mention in your reasoning:
(a) Why current evidence is sufficient.
(b) Why additional unseen frames are unlikely to alter your conclusion.
- If there's any reasonable scenario where unseen frames could alter your answer, you must explicitly acknowledge that and keep your confidence at "medium".

**Follow these instructions strictly.**

---

---

**Prompt Template for Stage II: Deep-dive Analysis and Refinement**

You are analyzing a video that is {duration_seconds} seconds long. The video has been uniformly sampled into 64 frames, indexed from 0 (start) to 63 (end).

Current context on question "{question}":

Current context in buffer "{buffer}".

Now analyze these {len(indices)} new frames (indices: {[int(idx) for idx in indices]}) from the gap ({start_idx}, {end_idx}).

**Tasks:**
- Provide a caption, relevance score (INTEGER 1-5) for each NEW frame, your UPDATED confidence, answer, and reasoning.
- If the new evidence changes your view of any PREVIOUS frame listed above, list the updated scores under "revised_prev_scores" (index, new relevance 1-5).

Respond in JSON: {{"new_frame_analysis": [{{"index": int, "caption": "str", "relevance": int}}], "revised_prev_scores": [{{"index": int, "relevance": int}}], "confidence": "str", "answer_attempt": "str", "reasoning": "str"}}

---

# D ADDITIONAL DATASET STATISTICS

To complement the main dataset description, we provide additional statistics that illustrate the textual and visual properties of FrameOracle-41K. Table 6 lists the 16 question types and their associated definitions, while Figure 6 visualizes several key quantitative aspects of the dataset.

Table 6: Question types and their corresponding definitions in FrameOracle-41K.

| Question type | Definition |
|---|---|
| Temporal | Designed to assess reasoning about temporal relationships between actions or events. Questions involve previous, present, or next actions. |
| Spatial | Tests ability to perceive spatial relationships between observed instances in a video scene. |
| Causal | Focuses on explaining actions or events and determining intentions, causes, or consequences. |
| Description Scene | Assesses ability to describe the major scene of the video, such as where it takes place and the overall environment. |
| Description Human | Involves describing actions or attributes of people, such as their activities and appearances. |
| Description Object | Assesses ability to describe attributes of objects, including appearance and function. |
| Count | Tests ability to count instances of objects, people, or actions, and to distinguish between old and new elements in a scene. |
| Binary | Involves yes/no questions related to the video content. |
| Fine-Grained Action | Creates questions that challenge comprehension of subtle or detailed actions. |
| Plot | Challenges ability to interpret the narrative or plot in the video. |
| Object Existence | Assesses reasoning with introduced non-existent activities while keeping physical scene details unchanged. |
| Time Order | Challenges recognition of the temporal sequence of activities in videos. |
| Object Direction | Emphasizes perception of object movement direction. |
| Camera Direction | Focuses on the direction of camera movement. |
| Speed | Delves into discerning variations in motion speed, including absolute and relative differences. |
| Attribute Change | Centers on how object or scene attributes change over time, such as size, shape, color, or other properties. |

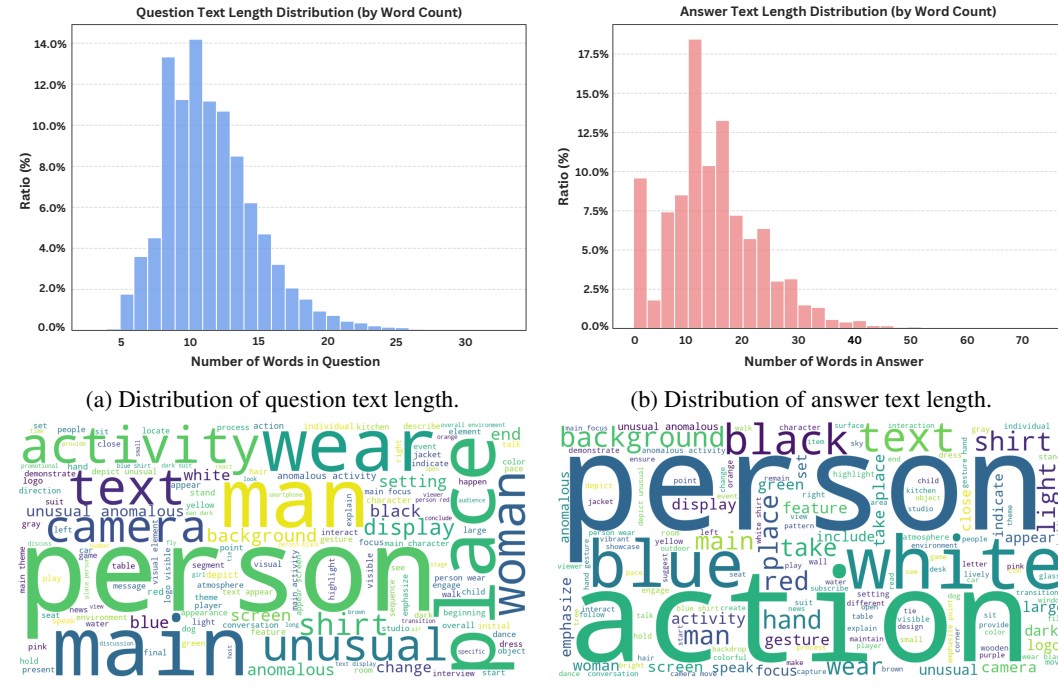

(a) Distribution of question text length.

(b) Distribution of answer text length.

(c) Word cloud of common terms in questions.

(d) Word cloud of common terms in answers.

Figure 6: **Additional dataset statistics.** (a) Distribution of question lengths. Most questions are short, typically ranging from 6 to 15 words. (b) Distribution of answer lengths. Answers show a slightly higher variance, indicating that some responses are longer or more descriptive. (c) Word cloud constructed from all questions, highlighting frequently used terms related to subjects, scenes, and activities. (d) Word cloud constructed from all answers, highlighting frequently used terms related to appearance, objects, and actions.

Figure 6a shows that most questions are short, typically between six and fifteen words, peaking around ten. This reflects our design goal of keeping each question focused on a single aspect of the video. Figure 6b shows answer lengths, which are generally similar to the questions but slightly more variable: most are short, yet some extend into longer, descriptive phrases. Together, these figures indicate that questions and answers are mostly compact, while answers allow some variation, supporting models in producing both concise labels and richer, sentence-level responses.

Figure 6c shows a word cloud of all questions. Frequent terms like person, place, activity, and wear indicate that questions often focus on subjects, scenes, and actions, reflecting diverse reasoning types in the dataset. Figure 6d shows a word cloud of all answers, where common words such as person, action, and various color terms highlight the focus on actions, appearances, objects, and scene details.

Together, these statistics provide a clear picture of FrameOracle's composition and design. Questions and answers remain concise yet varied, capturing both factual and descriptive reasoning. Videos provide sufficient temporal coverage without redundancy, and the lexical patterns demonstrate a balanced emphasis on visual grounding and action understanding.

## E   HUMAN VERIFICATION OF FRAMEORACLE-41K ANNOTATIONS

To ensure the reliability of FrameOracle-41K's automatically generated annotations, we conduct a human verification study on 4,000 randomly sampled instances ($\approx 10\%$ of the dataset). Ten independent annotators participate, with each sample reviewed by two distinct annotators. A sample is considered correct only if both annotators can answer the question using only the provided keyframes, without access to the full video or ground-truth answer. Inter-annotator agreement is high, at 94%, indicating strong consistency. As shown in Table 7, the overall human-verified accuracy is 93.3%

Table 7: Distribution of pairwise annotation outcomes on the human verification set (4,000 samples). A sample is classified as "Both correct" if both annotators answered the question correctly using the provided keyframes, "One correct / one wrong" if exactly one annotator answered correctly, and "Both wrong" if neither annotator answered correctly.

| Annotator Agreement Category | Count | Percentage |
| --- | --- | --- |
| Both correct | 3,732 | 93.3% |
| One correct / one wrong | 240 | 6.0% |
| Both wrong | 28 | 0.7% |

Table 8: **Human verification on frame sufficiency.** We classify the annotated keyframe sets into three categories based on the gap between the annotated count and the human-perceived minimal sufficiency.

| Category | Criterion | Count | Percentage |
| --- | --- | --- | --- |
| Excessive | $\geq 5$ frames beyond minimal sufficiency | 3,628 | 97.2% |
| Just-right | within $\pm 4$ frames of minimal sufficiency | 98 | 2.6% |
| Insufficient | Missing critical evidence | 6 | 0.2% |

(3,732/4,000), confirming that the vast majority of mined keyframe sets provide sufficient evidence for human-level reasoning.

Beyond correctness, we assess whether the annotated frames follow the principle of "minimal sufficiency." For all verified samples, annotators label keyframe sets as *Excessive*, *Just-right*, or *Insufficient*. Table 8 provides detailed definitions and statistics. As shown, the vast majority of samples (97.2%) are *Just-right*, with only a small fraction classified as excessive or insufficient. This confirms that FrameOracle-41K provides high-quality supervision that is both semantically accurate and frame-efficient, validating the effectiveness of our automated multi-stage generation pipeline.

## F  ADDITIONAL ABLATION STUDIES AND ANALYSIS

In this section, we present ablation studies and comparative analyses to validate FrameOracle's architectural and methodological choices. We examine six key aspects to highlight the robustness and efficiency of our approach: (1) the **necessity of the multi-stage training curriculum**, showing that intermediate weak supervision is crucial for avoiding overfitting and learning generalized policies; (2) the **impact of the supervision backbone**, demonstrating that the selector generalizes across different teacher models and effectively leverages stronger visual representations to enhance long-context reasoning; (3) the **source of performance gains**, confirming through equal-budget comparisons that improvements come from selecting semantically relevant frames rather than simply reducing visual input; (4) the **stability of freezing the Rank Head**, verifying that ranking consistency is preserved in Stage 3 and fully refined by subsequent fine-tuning in Stage 4; (5) the benefits of **frame-level over token-level budgeting**, demonstrating that maintaining whole-frame integrity outperforms rigid token limits; and (6) the advantage of **explicit selection over memory compression**, with substantial gains over long-video compression baselines. Together, these results show that FrameOracle's effectiveness stems from intelligent, query-conditioned semantic frame selection.

### F.1  NECESSITY OF THE MULTI-STAGE TRAINING CURRICULUM

Our training pipeline adopts a specific four-stage curriculum. To validate this design, we conduct two complementary ablation studies using the Qwen2.5-VL-3B backbone: (1) investigating whether intermediate weak supervision stages can be skipped (Stage 1+4), and (2) examining whether they can be merged into a single joint optimization step (Joint Training).

**Can we skip the intermediate stages?** To assess whether the four-stage curriculum is necessary or if the model could learn solely from Stage 4's strong supervision (FrameOracle-41K), we perform an ablation that compare the full pipeline against a simplified version using only Stage 1 (Text–Visual

Table 9: **Ablation of training stages.** Results are reported on Qwen2.5-VL-3B with 32 candidate frames. Average accuracy is calculated across all benchmarks listed in Table 1.

| Setting | Frame Selection | Avgerage Accuracy |
|---|---|---|
| Qwen2.5-VL-3B (Baseline) | 32 | 50.5 |
| + Stage 1 | $32 \to 16$ | 48.5 |
| + Stage 1+4 (Fixed K) | $32 \to 16$ | 49.6 |
| + Stage 1+2 | $32 \to 16$ | 50.8 |
| + Stage 1+4 (Adaptive K) | $32 \to 13.4$ | 46.9 |
| **+ Full (Stage 1–4)** | $\mathbf{32 \to 20.9}$ | **52.8** |

Table 10: **Joint Training vs. Staged Training.** Comparison of downstream performance with a fixed 16-frame budget.

| Training Strategy | Frames | NExTQA | | | Perception | LVB | Video-MME | EgoSchema | MLVU |
|---|---|---|---|---|---|---|---|---|---|
| | | OE_val | OE_test | MC | | | | | |
| Stage 2+3 (joint) | $32 \to 16$ | 24.0 | 27.8 | 72.2 | 63.8 | 50.1 | 54.3 | 49.8 | 52.2 |
| Stage 2 alone | $32 \to 16$ | **24.8** | **29.5** | **73.0** | **64.7** | **51.9** | **55.7** | **52.2** | **54.8** |

Alignment) and Stage 4 (Supervised Fine-tuning), evaluating two settings: (1) fixed $K$ and (2) adaptive $K$ prediction. As Table 9 shows, skipping intermediate stages reduces performance. Stage 1+4 improves over Stage 1 alone but still falls short of the Stage 1+2 baseline. Notably, enabling the K Head without Stage 3 calibration (Adaptive K) drops average accuracy to 46.9% and causes overfitting to FrameOracle-41K's statistics. These results confirm that the weak supervision in Stages 2 and 3 is crucial for learning generalized ranking and frame-budget policies before refinement with ground-truth annotations.

**Can we jointly train Stage 2 and 3?** We further investigate whether Stage 2 (Rank optimization) and Stage 3 (K optimization) could be simplified into a single joint training phase. We train a variant where both heads are optimized simultaneously. We observe severe optimization instability under the joint setting. Specifically, the K Head collapses, predicting near-maximum frames (i.e., failing to perform meaningful frame reduction), and the Rank Head becomes unstable, with Kendall-$\tau$ fluctuating between -0.4 and +0.6. This instability arises because the two objectives interfere during joint optimization: immature Rank predictions generate noisy top-K subsets that destabilize K-Head learning, and the resulting unstable K outputs further corrupt Rank learning, creating a feedback loop that prevents either head from converging. To quantify the impact, we measure the ranking consistency (Kendall-$\tau$) against ground truth and evaluate downstream performance under a fixed 16-frame budget (to isolate ranking quality).

As shown in Table 10, the jointly trained model achieves a Kendall-$\tau$ of only **0.2313** (vs. 0.5367 for Stage 2 alone) and consistently underperforms the staged baseline across all benchmarks. These results confirm that decoupling the ranking and budgeting objectives via a curriculum is essential for stable optimization.

## F.2 Impact of the Supervision Backbone

In our default setting, Stages 2 and 3 use Qwen2.5-VL-3B (Bai et al., 2025) to provide soft supervision (via VLM loss) for training the Rank Head and K Head. To assess whether FrameOracle relies on specific architectural biases of the teacher model, we conduct an ablation by replacing Qwen2.5-VL-3B with the more powerful VideoLLaMA3-7B (Zhang et al., 2025a) during training. To ensure a controlled comparison, both selector variants are evaluated using the same downstream pipeline (LLaVA-OneVision-7B (Li et al., 2025)) on the same benchmarks.

As shown in Table 11, replacing the backbone has minimal effect on the predicted frame counts, with both selectors producing nearly identical values (13.9 versus 14.2). Crucially, both variants outperform the full-frame baseline, demonstrating that FrameOracle generalizes well across different visual backbones. Furthermore, the selector trained with VideoLLaMA3-7B achieves stronger performance across benchmarks. This improvement stems from VideoLLaMA3-7B's stronger vi-

Table 11: Comparison of FrameOracle models trained with different VLM backbones (Stage 2/3).

| Model | Frames | NExTQA | | | Perception | LVB | Video-MME | EgoSchema | MLVU |
|---|---|---|---|---|---|---|---|---|---|
| | | OE_val | OE_test | MC | | | | | |
| LLaVA-OneVision-7B | 16 | 14.6 | 16.7 | 78.2 | 56.4 | 55.0 | 56.1 | 60.8 | 60.9 |
| + FrameOracle (VideoLLaMA3-7B) | 13.9 | 16.5 | **19.0** | **78.5** | 56.9 | 56.5 | 58.1 | 63.4 | 63.7 |
| + FrameOracle (Qwen2.5-VL-3B) | 14.2 | **16.7** | 18.9 | **78.5** | **57.0** | **57.4** | **58.4** | **64.0** | **65.1** |

Table 12: Evaluation on LLaVA-OneVision-7B under an equal frame budget.

| Model | Frames | NExTQA | | | Perception | LVB | Video-MME | EgoSchema | MLVU | Avg. |
|---|---|---|---|---|---|---|---|---|---|---|
| | | OE_val | OE_test | MC | | | | | | |
| LLaVA-OneVision-7B | $16 \rightarrow 10$ (Uniform) | 8.8 | 11.5 | 73.5 | 53.8 | 53.2 | 54.1 | 60.0 | 55.8 | 46.3 |
| + FrameOracle | $\mathbf{16 \rightarrow 10.4}$ | **16.1** | **17.8** | **77.6** | **56.5** | **55.5** | **56.0** | **62.4** | **60.2** | **50.3** |

sual representations, which provide richer information about events spanning longer periods; as a result, this selector is better at choosing frames that capture high-level or long-range context, leading to larger gains on long-video benchmarks (e.g., MLVU).

### F.3 DO FRAMEORACLE'S GAINS COME FROM FEWER FRAMES OR BETTER FRAMES?

We assess whether FrameOracle's improvement over the full-frame baseline comes from better frame selection rather than simply using fewer frames. To do this, we compare FrameOracle with a uniform sampling baseline under the same frame budget. Using LLaVA-OneVision-7B, Frame-Oracle reduces 16 input frames to an average of 10.4. We then uniformly sample 10 frames from the same 16-frame inputs and evaluate both on the same benchmarks. As Table 12 shows, uniform sampling achieves 46.3% average accuracy, while FrameOracle reaches 50.3%. This 4.0-point gain confirms that the improvement comes from selecting semantically relevant, query-conditioned frames, not merely from reducing the number of frames.

### F.4 EFFECT OF FREEZING THE RANK HEAD IN STAGE 3

We evaluate how freezing the Rank Head during Stage 3, while updating the Transformer encoder layers, affects its alignment with the evolving frame representations. This alignment is crucial for accurately ranking frames, especially in long videos where subtle differences in importance matter. To assess this, we perform a dedicated ablation measuring both (1) ranking consistency and (2) downstream task performance.

**Frame-Importance Consistency.** We randomly sample 500 training videos and compute a "ground-truth" importance distribution using leave-one-out (LOO) VLM loss. We then evaluate how well the Rank Head from Stages 2, 3, and 4 aligns with this distribution using Kendall-$\tau$ correlation. A large drop from Stage 2 to Stage 3 would indicate that freezing the Rank Head reduces its ability to track the encoder's evolving representations. As Table 13 shows, $\tau$ decreases only slightly after Stage 3. This is largely due to the very low learning rate ($1 \times 10^{-7}$) used for encoder updates, which limits feature drift. Stage 4 fully restores and even improves the alignment, demonstrating that any temporary misalignment is easily corrected through supervised fine-tuning.

**Downstream Benchmark Evaluation.** We assess the practical impact of freezing the Rank Head in Stage 3 by evaluating Stages 2, 3, and 4 on video benchmarks using the Qwen2.5-VL-3B backbone with 32-frame inputs. To ensure a controlled comparison, we fix the number of selected frames to 16 across all stages, isolating the effect of ranking drift. As shown in Table 14, Stage 3 performs on par with Stage 2, indicating that the temporary freeze causes only negligible drift. Stage 4 consistently boosts performance across all benchmarks, confirming that supervised fine-tuning effectively realigns and strengthens the ranking behavior. These results demonstrate that temporarily freezing the Rank Head does not meaningfully harm the selector, even in long-video scenarios where ranking stability is critical.

Table 13: Kendall-$\tau$ consistency across training stages.

| Model | Kendall-$\tau$ (vs. Ground Truth) |
|---|---|
| Stage 2 | 0.5367 |
| Stage 3 | 0.5221 |
| Stage 4 | **0.5833** |

Table 14: Downstream performance across training stages under a fixed 16-frame selection.

| Model | Frames | NExTQA | | | Perception | LVB | Video-MME | EgoSchema | MLVU | Avg. |
|---|---|---|---|---|---|---|---|---|---|---|
| | | OE_val | OE_test | MC | | | | | | |
| Stage 2 | 32→16 | 24.8 | 29.5 | 73.0 | 64.7 | 51.9 | 55.7 | 52.2 | 54.8 | 50.8 |
| Stage 3 | 32→16 | 24.8 | 29.5 | 72.7 | 64.9 | 52.0 | 56.0 | 51.8 | 54.4 | 50.8 |
| Stage 4 | 32→16 | **25.3** | **30.0** | **74.0** | **65.8** | **52.9** | **56.9** | **52.8** | **55.5** | **51.7** |

## F.5 FRAME-LEVEL VS. TOKEN-LEVEL INFORMATION BUDGETING

An important question in adaptive video understanding is how to define the unit of computational budget: by visual tokens or by frames. To investigate this, we compare two representative strategies:

- **Token-level Budgeting:** Operates at a fine-grained level, selecting specific spatial-temporal tokens to fit a **fixed** total budget. This may discard parts of a frame to save computation.
- **Frame-level Budgeting (Ours):** Operates at a coarser granularity, treating each frame as an atomic unit. It **dynamically predicts** a budget of $K$ full frames, preserving the complete spatial context of each selected frame.

To compare these strategies, we select B-VLLM (Lu et al., 2025) as a representative token-level budgeting method. B-VLLM samples videos at 1 fps and enforces a fixed budget of 512 visual tokens, selecting the most relevant spatial tokens across the sequence. We compare this against FrameOracle, which predicts a dynamic number of frames. For a fair comparison, we evaluate both methods under a unified backbone (VideoLLaMA2 (Cheng et al., 2024)) to isolate the effect of the selection mechanism. Additionally, we compare B-VLLM's best-reported performance with our LLaVA-Video integration, both using Qwen2 (Yang et al., 2024) as the backbone LLM. As shown in Table 15, FrameOracle consistently outperforms B-VLLM across all benchmarks. These results indicate that preserving the spatial integrity of frames is crucial for reasoning and that dynamically selecting the number of frames is a more effective approach for controlling visual information than enforcing a fixed token budget.

## F.6 COMPARISON WITH LONG VIDEO COMPRESSION METHODS

We further compare FrameOracle with memory-based compression methods designed for long video understanding. Unlike frame-selection approaches, methods such as MovieChat (Song et al., 2024), MovieChat+ (Song et al., 2025), and ReWind (Diko et al., 2025) process video streams sequentially, maintaining a learnable or fixed-size memory buffer that compresses historical frame features to control context length. For a fair comparison, we reproduce MovieChat using the same backbone as FrameOracle (LLaVA-OneVision). Following the official implementation, the reproduced MovieChat processes up to 512 input frames, compressing them into a maximum of 64 frames for the downstream VLM. We use the 64-frame version of FrameOracle for this comparison.

Table 16 summarizes the results on the MovieChat-1K benchmark and other standard long-video benchmarks. For reference, we also report the originally published numbers for MovieChat, MovieChat+, and ReWind. On MovieChat-1K, FrameOracle achieves 69.6% accuracy, showing that adaptive keyframe selection can match or surpass dense-frame baselines while using far fewer frames. On long-video benchmarks such as LongVideoBench, VideoMME, and EgoSchema, Frame-

Table 15: Comparison of information budgeting strategies: Frame-level (FrameOracle) vs. Token-level (B-VLLM (Lu et al., 2025)).

| Model | Information | Perception | VideoMME | EgoSchema |
|---|---|---|---|---|
| VideoLLaMA2+B-VLLM | 1fps → 512 tokens | 48.0 | 44.4 | 44.3 |
| VideoLLaMA2+FrameOracle | 64 → 12.9 frames | **53.4** | **54.1** | **51.8** |
| B-VLLM | 1fps → 512 tokens | 52.1 | 53.5 | 51.9 |
| LLaVA-Video+FrameOracle | 64 → 12.9 frames | **65.1** | **61.6** | **55.2** |

Table 16: **Comparison with long video compression methods.** Note that reference methods do not report results on newer benchmarks (indicated by -). Best performance of reference methods is indicated by underline.

| Model | Frame | MovieChat-1K | | LVB | VideoMME | EgoSchema |
|---|---|---|---|---|---|---|
| | | Acc. | Score | | | |
| *(1) Reference Methods* | | | | | | |
| MovieChat (Song et al., 2024) | 2048 | 62.3 | 3.81 | - | - | - |
| MovieChat+ (Song et al., 2025) | 2048 | 71.2 | 3.51 | - | - | - |
| ReWind (Diko et al., 2025) | 548 | 80.6 | 4.46 | - | - | - |
| *(2) Controlled Comparison (Backbone: LLaVA-OneVision)* | | | | | | |
| + MovieChat | 512 → 64 | 67.1 | 3.44 | 44.2 | 45.6 | 57.8 |
| **+ FrameOracle (Ours)** | **64 → 15.6** | **69.6** | **3.82** | **56.5** | **58.1** | **63.4** |

Oracle consistently outperforms compression-based methods, improving accuracy by up to +12.3% on LongVideoBench. These results indicate that adaptively selecting a small set of semantically rich keyframes provides stronger supervision and better generalization than compressing long frame sequences into fixed tokens or memory buffers.

# G ADDITIONAL BENCHMARKS DETAILS

Table 17 summarizes the evaluation prompts for each benchmark used in our experiments, most of which are adapted from LMMs-Eval.

Table 17: Prompts specifying the response format used for each evaluation benchmark.

| Benchmark | Response formatting prompts |
|---|---|
| MLVU | – |
| Video-MME | Answer with the option's letter from the given choices directly. |
| EgoSchema | Answer with the option's letter from the given choices directly. |
| NExTQA | – |
| Perception | Answer with the option's letter from the given choices directly. |
| LongVideoBench | Answer with the option's letter from the given choices directly. |

# H FRAMEORACLE-41K EXAMPLES

Figure 7 shows three examples from the FrameOracle-41K dataset. Each example demonstrates the number of selected keyframes, the associated question, the ground-truth answer, and the question type. We also provide indices for the selected keyframes within the 64 uniformly sampled frames used during preprocessing, indicating their relative positions along the video timeline. These examples highlight the diversity of reasoning types in FrameOracle-41K, such as causal reasoning and fine-grained action understanding, and illustrate that the annotations focus on semantically informative moments rather than evenly spaced frames.

**"duration":** 163.9638,
**"num_selected_frames":** 8,
**"selected_frames_index":** ["0", "12", "18", "24", "28", "29", "30", "31"],
**"question":** "Why does the video show an animated diagram of the solar system?",
**"ground_truth_answer":** "The animated diagram of the solar system is shown to explain the orbits of Mercury, Venus, Earth, Mars, and the comet 109P/Swift-Tuttle, which spawns the Perseids meteor shower.",
**"question_type":** "Causal"

**"duration":** 176.2761,
 **"num_selected_frames":** 20,
 **"selected_frames_index":** ["18", "19", "20", "32", "33", "34", "37", "38", "39", "41", "42", "43", "44", "45", "46", "47", "48", "52", "54", "57"],
**"question":** "How does the person demonstrate their skill and precision in the video?",
**"ground_truth_answer":** "The person demonstrates skill and precision by carefully manipulating metal with tongs, shaping it on a grinding wheel, and using a cordless drill to create precise holes in materials.",
**"question_type":** "Fine-grain Action"

**"duration":** 129.7630,
**"num_selected_frames":** 6,
**"selected_frames_inedx":** ["0", "6", "12", "18", "24", "41"],
**"question":** "Where does the video take place?",
**"ground_truth_answer":** "The video takes place in an indoor training facility.",
**"question_type":** "Description Scene"

Figure 7: **Examples from the FrameOracle-41K dataset.** Each example shows the number of selected keyframes, question, ground-truth answer, question type, and the indices of the selected keyframes.

## I  QUALITATIVE EXAMPLES

As shown in Figure 8, FrameOracle can achieve correct answers using far fewer frames than uniform sampling. In the illustrated examples, our selector retains only 2–4 frames out of the original 16 inputs, yet these frames provide sufficient evidence to answer the questions accurately. This highlights that many uniformly sampled frames are redundant and that FrameOracle effectively filters them without sacrificing accuracy.

Figure 9 presents cases related to RQ1 (Section 5.2). Providing all 16 uniformly sampled frames can introduce irrelevant or distracting content, leading the VLM to produce incorrect answers. In contrast, FrameOracle selects a smaller, query-focused subset, allowing the model to concentrate on relevant evidence and answer correctly. These examples illustrate that more frames do not necessarily improve performance, and adaptive selection of fewer, informative frames enhances understanding.

*Question*: What did the man in the front do when the man at the back after the man at the back picked up the spoon?

A. take the bowl                                        B. places pan back on stove
**C. dip food into sauce**                      D. does hand gesture toward the tiger
E. help to season other chickens

**Uniform Sampling (Qwen2.5-VL: C)**

**FrameOracle (Qwen2.5-VL: C)**

*Question*: Why is the man wearing slippers sitting at the top of the rock at the start of the video?

A. take photo from angle                       B. sunbathing
C. preparing for a performance           D. to pose for the camera
**E. waiting to jump in water**

**Uniform Sampling (Qwen2.5-VL: E)**

**FrameOracle (Qwen2.5-VL: E)**

Figure 8: **Qualitative examples.** FrameOracle answers correctly while using only a few frames (2 to 4 out of 16), compared to uniform sampling, which relies on the full input.

*Question*: What seems to be the main purpose of the video? What actions did c perform to achieve this purpose?

A: The main objective of this instructional video is to effectively demonstrate how to easily tie your hair back.
B: The main purpose of the video is to show how to open a jar.
C: The primary objective of the video presentation is to demonstrate the most effective methods for properly cleaning your windows.
**D: The main purpose of the video is to show how to use a resistance band to exercise your arms and upper body.**
E: The primary objective of this video presentation is to effectively demonstrate the proper way to engage in a fun tug-of-war match with your canine companion.

**Uniform Sampling (Qwen2.5-VL: A)**

**FrameOracle (Qwen2.5-VL: D)**

*Question*: From the sequence of actions, identify a turning point or moment where c's focus shifts to a different task. explain why you believe this is the most significant part of the video.

**A: The turning point is when c unfastens the hub axle.**
B: The crucial turning point occurs when character c picks up the screwdriver from the table.
C: The pivotal turning point occurs when character c decides to put on the gloves.
D: The turning point is when c removes the tire.
E: The critical turning point occurs when character c successfully patches the hole, fixing it.

**Uniform Sampling (Qwen2.5-VL: B)**

**FrameOracle (Qwen2.5-VL: A)**

Figure 9: **Qualitative examples for RQ1.** Using all 16 uniformly sampled frames can produce incorrect answers, whereas FrameOracle answers correctly by selecting only the relevant subset.

