# OpenReview forum: "FrameOracle: Learning What to See and How Much to See in Videos"
_ICLR.cc/2026/Conference — Submitted to ICLR 2026_

### Official Review · Reviewer_p9ow · 2025-10-15

**Soundness:** 2
**Presentation:** 3
**Contribution:** 2
**Rating:** 2
**Confidence:** 4

**Summary:**

The paper addresses a key challenge in video understanding with vision-language models (VLMs):
How to efficiently select the most relevant frames from a video to answer a given query, while also determining how many frames are actually needed. To answer these questions the authors propose a 'plug-and-play' frame selection (named FrameOracle) to select which and how many frames to use for a given query. In addition, the paper also contributes a dataset named FrameOracle-41K which contains information regarding the important frames for each query inside the dataset.

**Strengths:**

1 - Adaptive frame selection method that selects how many frames and which frames to use, an extension of what current SOTA models are doing.
2 - New dataset: the dataset is a significant contribution of this work, despite the little attention given to it.
3 - The authors have considered many different datasets which increase the reliability of the method in the considered settings.

**Weaknesses:**

After carefully reading the paper I have the following doubts which I categorize as weaknesses:

1 - The adaptive frame selection setting: While the authors claim adaptive frame selection, they are in fact doing adaptive frame selection from a fixed pool of frames. This is not aligned with the nature of videos which can be in variable sizes. This becomes more critical when you apply the method to long video understanding, depending on the quantity of information the video contains and the dynamic, using 16 or 64 frames (uniformly sampled before the frame selection mechanism) is not enough let alone when reduced to ~10 frames. While this is mentioned as a limitation from the authors, I fail to recognize an important contribution from the method if the frame selection cannot operate on variable sequence length (or even fixed but complete) since it is always bounded to the correctness of the uniform sampling.

2 - Plug and play: The method is not actually plug and play since the query is encoded from the vlm tokenizer, which means for every VLM with different tokenizer, you have to train a separate model. It would have been plug and play if you would have used the for example the SigLIP language encoder.

3 - Feature fusion: the cross-modal fusion is an integral part of your technical contributions, but it is unclear how you do it. Do you concatenate the tokens?

4 - Transformer encoder layer: Features are fused and then sent to the transformer encoder layer. How does the transformer process the tokens, is it a global attention (i.e. all tokens from all frames), is it spatial and then temporal, or is it only spatial? This is related to weakness 3 also. I guess the architecture part is a bit undermined in this work. The reason why I stress this point is because the frame selection is mainly a mechanism to reduce computation while keeping or improving the accuracy. In terms of memory, for an LLM/VLM the most expensive operation is the self-attention (let's consider a plain self-attention) due to quadratic memory scaling. Now, if the encoder layer is using self-attention among all the tokens, it means the transformer encoder has memory requirements similar to those of the LLM/VLM during computations (the difference would be the number of heads) with exception to the system prompt tokens. So, while reducing latency, the frame selector has big memory requirements. I suggest this point is clarified.

5 - Why not use the visual tokenizer of the VLM directly, the method is not plug-and-play anyway. The visual tokenizer of the VLM is already align with language (so no stage-1 training), and can possibly ease the 'which frames to use' problem. (Note, this will not affect my evaluation negatively, is more for my own curiosity.)

6 - The 4 stage training: What would happen if the training would consist only of stage-1 and 4? An experiment would be interesting. Additionally, stage-2 and -3 freeze and unfreeze the Rank head and K head. If you train the model (even with a low learning rate) in stage-3 while you freeze Rank head, the performance of Rank head will decrease. Now how big of the problem this is depends on the frames given in input and the video composition and it would be problematic for long videos and scenarios with high sensitivity on the frame choice. The approach is not validated rigorously to have a conclusion on this matter.

7 - Validation: While the paper validates across many datasets and compares with previous works in different datasets, I think it is evaluated in a very shallow way without depth. The ablations are not very comprehensive, just the 4 stages. No insights on the visual backbone choice or text tokenizer, and many more (see above).

8 - Dataset: The dataset, in my understanding would be the bigger contribution to this work but it is clearly overlooked and very little analysis and experiments are done with it.

9 - While the work considers only frame selection mechanism, entering in the long video world, I guess is fair to consider and compare against works that apply to long video understanding (is not necessary for the proposed method to surpass those works, but to have an idea how it actually helps when compared to methods designed for long videos).
You can have a look at: "Moviechat+: Question-aware sparse memory for long video question answering." IEEE Transactions on Pattern Analysis and Machine Intelligence (2025), which is training free but uses a memory layer (plug-and-play) to compress frames rather than select them, or "ReWind: Understanding Long Videos with Instructed Learnable Memory." Proceedings of the Computer Vision and Pattern Recognition Conference (2025). This is more a suggestion to see how selecting compares to compression.


Given these concerns, I will suggest a weak reject. The work has no significant technical contributions, it is incremental. In addition, the theoretical contributions are not significant to compensate the technical ones. The dataset is the only significant contribution but is clearly not the main focus of the paper. I am open to improve my score if my concerns are addresses or my interpretations are deemed as not correct.

**Questions:**

Check the Weaknesses section.

---

> ### Author Response · Authors · 2025-11-21
> **Response to Reviewer p9ow (1)**
>
> Thank you for your detailed feedback and for recognizing the significance of our FrameOracle-41K dataset and our thorough evaluation demonstrating the reliability of our approach.
>
> > **The adaptive frame selection setting ...**
>
> * **On adaptive selection**: “Adaptive” means the model decides both which frames to keep and how many are needed based on the video and query. Although we input a fixed-size candidate pool (e.g., 16 or 64 frames), FrameOracle selects a variable number of frames, unlike fixed-budget strategies that always select the same number (e.g., Frame-Voyager, BOLT, KFC). This enables content- and query-aware selection. Moreover, our experiments show that using a fixed 64-frame candidate pool is not universally insufficient. As reported in Table 1, FrameOracle applied on 64 uniformly sampled candidates achieves consistent improvements over strong VLM baselines across diverse long-video benchmarks (e.g., LongVideoBench, VideoMME, EgoSchema, MLVU).
>
> * **On uniform sampling:** We agree that uniform sampling is limited, but FrameOracle is not restricted by the upper bound of it. As shown in Table 1 and Figure 9 (Appendix), and discussed in RQ1 (Sec. 5.1), FrameOracle achieves higher performance on multiple benchmarks while selecting fewer frames from the same 16 candidates, demonstrating its ability to identify key evidence and remove the redundant frames that could hurt the reasoning accuracy by introducing visual noise and diluting the model's attention.
>
> * **On variable-length inputs:** We experimented with variable-length inputs (e.g., 1 fps), but in Stages 2 and 3, large variations in frame count led to unstable training and non-convergence due to unnormalized supervision signals and noisy gradients. To maintain stable training, we use a fixed candidate pool. We rely on VLM loss as soft supervision because large-scale datasets with frame-level relevance labels do not yet exist. While FrameOracle-41K provides such annotations, its size makes it more suitable for fine-tuning than for pretraining.
>
> There are two natural paths to scale FrameOracle further: (1) expand FrameOracle-41K to a much larger dataset to enable fully supervised training of the Rank and K Heads, though this is currently constrained by annotation cost, and (2) increase the fixed candidate pool from 64 to 256 or 512 frames to approximate full long-video coverage (~1 fps) while preserving the same training framework. Both directions are promising avenues for future work.
>
> > **Plug and play: The method is not actually plug and play ...**
>
> FrameOracle is fully plug-and-play for inference across diverse VLMs.
> The confusion arises from wording in the original version L243–244: “encoded using the native tokenizer of the downstream VLM” referred only to how text embeddings were obtained during *training*, not to any architectural dependence on a specific tokenizer. We have corrected the wording in our revision. As clarified in our revision, Section. 5.1 (L351–352), both FrameOracle models were trained with the Qwen2.5-VL tokenizer for consistency, but at inference they operate solely on text embeddings output by the text projector, making the system tokenizer-agnostic. Tables 1 and 2 further show that the same FrameOracle model, trained once with the Qwen2.5-VL tokenizer, was applied directly, without retraining, to multiple VLMs (LLaVA-OneVision, LLaVA-Video, VideoLLaMA, GPT-4o, etc.), consistently improving performance.
>
> > **Feature fusion: Do you concatenate the tokens?**
>
> Cross-modal fusion is implemented by **concatenating** projected text and frame tokens. After projecting both modalities into a shared space, we prepend a learnable query token to form $[k_{\text{query}}; t_{\text{text}}; f_{\text{frame}}]$, which is processed by a shared Transformer encoder where self-attention enables token-level cross-modal interaction. We also add modality-type and positional embeddings to distinguish text and frame tokens, and the shared self-attention performs symmetric bidirectional fusion (rather than one-way cross-attention).
>
> > **Transformer encoder layer**
>
> Each frame in FrameOracle is represented by **a single high-level embedding** from a pretrained visual backbone (DINOv2), analogous to a [CLS] token summarizing the frame. One frame token thus corresponds to the entire frame, and the encoder operates on **frame-level tokens** along with a few text tokens and one learnable query token.
>
> The encoder uses global self-attention over this short sequence for temporal and cross-modal reasoning. It is lightweight (4 layers, 8 heads, 256 hidden size) and models relationships among frame embeddings rather than pixel- or patch-level features. The attention is fully global across all frame and text tokens, not decomposed into spatial or temporal stages. As Table 3 shows, this adds less than 1% to the computation and memory cost of the downstream VLM. We will clarify these details in the final version.

---

> > ### Author Response · Authors · 2025-11-21
> > **Response to Reviewer p9ow (2)**
> >
> > > **Why not use the visual tokenizer of the VLM directly, the method is not plug-and-play anyway...**
> >
> > FrameOracle is fully plug-and-play. Once trained, it can attach to any downstream VLM with no retraining or architectural changes (see above). This universality comes from using a standalone visual encoder (DINOv2) that outputs frame embeddings with consistent dimensionality across models. Replacing it with the downstream VLM’s native visual encoder would necessitate retraining FrameOracle for every new target model, fundamentally breaking this "plug-and-play" efficiency. Alternatively, suppose we were to adopt a specific fixed VLM encoder as our visual encoder. In that case, we face a different issue: since state-of-the-art VLMs typically fine-tune their visual encoders jointly with the backbone LLM ([1],[2]), using such an encoder would cause our selector to overfit to that specific VLM’s biases rather than learning to objectively identify the frames that contain the answer.
> >
> > > **What would happen if the training would consist only of stage 1 and 4?**
> >
> > We conducted an ablation on **Stage 1+4 only**, under (1) **fixed-frame** and (2) **adaptive frame** settings:
> >
> > | Setting| Frame| Avg. Acc. |
> > |-----------------|---------|-------|
> > |Qwen2.5-VL-3B| 32|50.5|
> > |+Stage1| 32→16| 48.5|
> > |+Stage1+4 (fixed K)| 32→16| 49.6|
> > |+Stage1+2| 32→16| 50.8|
> > |+Stage1+4 (adaptive K) | 32→13.4 | 46.9|
> > |+Full (1–4)| 32→20.9 |**52.8**|
> >
> > * **Fixed K setting**: Stage 1+4 improves slightly over Stage 1 (+1.1%) due to supervised refinement from FrameOracle-41K, but still underperforms Stage 1+2 (–1.2%) despite using ground-truth labels. This highlights the importance of the curriculum: Stage 2 leverages ~300K samples with soft supervision to establish a broad, generalized ranking prior (acting as a "warm-up"), while FrameOracle-41K provides high-quality but smaller-scale (41K) strong supervision. Without the generalized policy learned from the large-scale data in Stage 2, the selector struggles to learn a robust ranking capability from scratch and tends to overfit the specific patterns of the 41K SFT data.
> >
> > * **Adaptive-K setting**: Enabling the K Head under Stage 1+4 drops performance to 46.9%, with predicted K collapsing to an average of 13.4 frames and most samples selecting 10–16 frames, mirroring FrameOracle-41K’s modal K distribution (after Qwen2.5-VL merges each two adjacent frames). Because Stage 1 does not train the K Head, Stage 4 becomes its only supervision, leading it to overfit dataset statistics rather than learn the accuracy–frame trade-off captured by Stage 3. This shows that supervised K labels alone are insufficient and that Stage 3 is essential for task-driven K calibration.
> >
> > These results support the curriculum design: Stages 2 and 3 are essential for stable ranking and frame-count prediction, while Stage 4 provides supervised refinement. Scaling and diversifying FrameOracle-41K could reduce the gap between Stage 1+4 and the full pipeline, potentially making Stages 2 and 3 unnecessary. We plan to explore this direction to develop a selector trained purely from supervised annotations.

---

> > > ### Author Response · Authors · 2025-11-21
> > > **Response to Reviewer p9ow (3)**
> > >
> > > > **Stage 2 and 3 freeze and unfreeze the Rank head and K head.**
> > >
> > > Freezing the Rank Head in Stage 3 could weaken its alignment with frame representations, but Stage 4 SFT is expected to correct this. To verify, we ran two experiments using the Qwen2.5-VL-3B backbone with 32-frame inputs: (1) **Frame-Importance Consistency Test:** We randomly sampled 500 training videos and computed ground-truth importance distributions using leave-one-out (LOO) VLM loss. We measured Kendall-τ correlations between these distributions and predictions from Stage 2, 3, and 4 to assess alignment. A drop from Stage 2 to 3 would indicate degradation, and recovery in Stage 4 would indicate correction. (2) **Benchmark Evaluation:** We evaluated Stage 2, 3, and 4 models on the same benchmarks using a fixed selection of 16 frames to measure downstream performance. The results appear in the tables below.
> > >
> > > | Model | Kendall-τ (vs. Ground Truth) |
> > > | :--- | :---: |
> > > | Stage 2 | 0.5367 |
> > > | Stage 3 | 0.5221 |
> > > | Stage 4 | 0.5833 |
> > >
> > >
> > > | Model | Frame | NExTQA_OE_val | NExTQA_OE_test | NExTQA_MC | Perception | LongVideoBench | VideoMME | EgoSchema | MLVU | Avg. Acc. |
> > > | :--- | :---: | :---: | :---: | :---: | :---: | :---: | :---: | :---: | :---: |  :---: |
> > > | Stage 2 | 32→16  | 24.8 | 29.5 | 73.0 | 64.7 | 51.9 | 55.7 | 52.2 | 54.8 | 50.8 |
> > > | Stage 3 | 32→16 | 24.8 | 29.5 | 72.7 | 64.9 | 52.0 | 56.0 | 51.8 | 54.4 | 50.8 |
> > > | Stage 4 | 32→16 | **25.3** | **30.0** | **74.0** | **65.8** | **52.9** | **56.9** | **52.8** | **55.5** | **51.7** |
> > >
> > > The results show that freezing the Rank Head in Stage 3 causes only **minimal feature drift**, which Stage 4 effectively corrects. This is mainly because using a very low learning rate (1e−7) on the encoder during Stage 3 limits deviations in the frame representations. Kendall-τ drops slightly from Stage 2 (0.5367) to Stage 3 (0.5221), but average downstream performance remains virtually unchanged, indicating negligible drift. Stage 4 corrects this misalignment and improves ranking quality, reaching **0.5833** Kendall-τ and **51.7%** average accuracy, demonstrating that the final SFT stage effectively realigns and strengthens the selection policy.
> > >
> > > > **Validation: The ablations are not comprehensive ...**
> > >
> > > We added several analyses: (1) an ablation on Stage 1+4 and another on freezing the Rank Head in Stage 3; (2) comparisons with long-video compression methods; and (3) a comparison with B-VLLM [3], which uses a fixed token budget instead of a frame-count budget.
> > >
> > > For (3), we compare frame-count budgeting (FrameOracle) with token budgeting (B-VLLM) as two strategies for controlling visual information. B-VLLM samples videos at 1 fps with a fixed 512-token visual budget. Its strongest performance is in the standalone setting with Qwen2, which we use to fairly compare against LLaVA-Video+FrameOracle, which also uses Qwen2. We also evaluate both methods under a unified backbone (VideoLLaMA2) to isolate the effect of the selection mechanism. In both settings, FrameOracle consistently outperforms B-VLLM, showing that learning a per-instance frame-count budget is an effective way for adaptive information control in video LLMs, without token-level merging or architectural changes.
> > >
> > > | Model | Perception | VideoMME | EgoSchema |
> > > | :--- | :---: | :---: | :---: |
> > > | VideoLLaMA2+B-VLLM | 48.0 | 44.4 | 44.3 |
> > > | VideoLLaMA2+FrameOracle | **53.4** | **54.1** | **51.8** |
> > > | B-VLLM | 52.1  | 53.5 | 51.9 |
> > > | LLaVA-Video+FrameOracle | **65.1** | **61.6** | **55.2** |
> > >
> > > We are also running an ablation using VideoLLaMA3 as the backbone in Stages 2 and 3 to study how backbone capacity affects FrameOracle. Results will be shared in this thread as soon as they are ready. All additional ablations and backbone comparisons are detailed in Appendix F of the revision.
> > >
> > > We hope these analyses address your question. If there are specific evaluations you would like to see, we would be happy to include them in the next rebuttal.

---

> > > > ### Author Response · Authors · 2025-11-21
> > > > **Response to Reviewer p9ow (4)**
> > > >
> > > > > **The dataset is clearly overlooked ...**
> > > >
> > > > We have expanded the dataset analysis and validation in the revised manuscript:
> > > >
> > > > (1) **Human verification**. We conducted a **human subject study** on 4,000 randomly sampled instances (~10% of the dataset). Ten annotators participated, each reviewing 800 samples, with two annotators per sample. A sample was considered *verified* only if both annotators correctly answered the question using the provided keyframes. The detailed result is shown in the table below. Inter-annotator agreement reaches 94%, indicating significant consistency. The overall human-verified accuracy is **93.3%** (3,732/4,000), confirming that most keyframe sets provide sufficient evidence for human reasoning.
> > > >
> > > > | Agreement Type     | Count | Percentage |
> > > > |--------------------|-------|-------------|
> > > > | Both correct       | 3,732 | 93.3% |
> > > > | One correct / one wrong | 240 | 6.0% |
> > > > | Both wrong         | 28 | 0.7% |
> > > >
> > > > For all verified samples, annotators further labeled each keyframe set as:
> > > > * Excessive: ≥5 frames beyond minimal sufficiency,
> > > > * Just-right: within ±4 frames of minimal sufficiency,
> > > > * Insufficient: missing critical evidence.
> > > >
> > > > The breakdown is: **97.2%** Just-right (3,628/3,732), **2.6%** Excessive (98/3,732), and **0.2%** Insufficient (6/3,732). These findings show that **most annotations are both semantically accurate and frame-efficient**, demonstrating that FrameOracle-41K provides high-quality supervision.
> > > >
> > > > (2) **Question-type and keyframe sufficiency analysis**. Using the taxonomy from LLaVA-Video-178K, we categorized all samples into 16 question types (e.g., Action, Object, Temporal, Spatial, Counting, etc.), and report both their prevalence and the distribution of minimal sufficient keyframes per type. This provides insight into the semantic variability of keyframe sufficiency across diverse question types. Full statistics are in Figure 3 in Section 3.2 and Table 6 in the revised Appendix D.
> > > >
> > > > (3) **Additional dataset statistics**. We added visualizations of question/answer length distributions and answer token-level word clouds to better characterize the dataset’s structure and linguistic diversity (Figure 6 in the revised Appendix D).
> > > >
> > > > We hope the revised version highlights the value and rigor of FrameOracle-41K.
> > > >
> > > > > **Compare against long video understanding methods ...**
> > > >
> > > > We added comparisons to long-video QA approaches (MovieChat [4], MovieChat+, Rewind) that use memory-based compression instead of frame selection.
> > > >
> > > > * **Evaluation on MovieChat-1K**. The table below reports results on MovieChat-1K. The last two rows use a unified backbone (LLaVA-OneVision) in a comparable inference-only setting: we reproduce MovieChat with its official implementation via Lmms-Eval (**up to 512-frame input, compressed to 64**) and apply FrameOracle to select an average of 15.6 frames from the same 64-frame input.
> > > >
> > > > FrameOracle outperforms MovieChat while using 8–32× fewer frames.
> > > >
> > > > | Model | Frame | MovieChat-1K-Global-Acc | MovieChat-1K-Global-Score |
> > > > | :--- | :---: | :---: | :---: |
> > > > | **(1) Reference Methods (Original Papers)** | | | |
> > > > | MovieChat | 2048 | 62.3 | 3.81 |
> > > > | MovieChat+ | 2048 | 71.2 | 3.51 |
> > > > | ReWind | 548 | 80.6 | 4.46 |
> > > > | **(2) Controlled Comparison (Backbone: LLaVA-OneVision)** | | | |
> > > > | + MovieChat (Reproduced) | 512 | 67.1 | 3.44 |
> > > > | **+ FrameOracle (Ours)** | **64→15.6** | **69.6** | **3.82** |
> > > >
> > > > * **Evaluation on additional long video benchmarks**. To assess generality across long-video scenarios, we compared LLaVA-OneVision+MovieChat with LLaVA-OneVision+FrameOracle on four standard long-video benchmarks. On all benchmarks, FrameOracle outperforms MovieChat while using far fewer frames:
> > > >
> > > > | Model | Frame | LongVideoBench | VideoMME | EgoSchema | MLVU
> > > > | :--- | :---: | :---: | :---: | :---: | :---: |
> > > > | LLaVA-OneVision+MovieChat | 512 | 44.2 | 45.6 | 57.8 | 47.7 |
> > > > | LLaVA-OneVision+FrameOracle | 64→14.8 | **56.5** | **58.1** | **63.4** | **63.7** |
> > > >
> > > > ---
> > > >
> > > > [1] VideoLLaMA3: Frontier Multimodal Foundation Models for Image and Video Understanding. arXiv'25.
> > > >
> > > > [2] Qwen2.5-VL Technical Report. arXiv'25.
> > > >
> > > > [3] B-VLLM: A Vision Large Language Model with Balanced Spatio-Temporal Tokens. ICCV'25.
> > > >
> > > > [4] MovieChat: From Dense Token to Sparse Memory for Long Video Understanding. CVPR'24.

---

> > > > > ### Comment · Reviewer_p9ow · 2025-11-21
> > > > > **Corrected version**
> > > > >
> > > > > I believe the work is now significantly more complete. I followed the authors' updates as they addressed the comments.
> > > > > The ablation study has become much more comprehensive, allowing readers to clearly understand the rationale behind each component. While the improvements are meaningful, they are not groundbreaking due to the inherent limitation on the number of frames. For instance, the authors note that with variable-length inputs, the models struggle to converge because of high variance—this remains a substantial limitation of the approach.
> > > > > However, the new experiments demonstrate that the method has merit, and I am raising my score accordingly. Regarding the comparison with MovieChat and Rewind: if you present the frames, it is important to also show the number of tokens consumed.
> > > > > This experiment is valuable because it highlights the method’s trade-offs: it achieves efficiency through frame selection, but this comes at the cost of temporal fidelity, as the sequence of events is altered. I appreciate the additional experiments provided. Please feel free to add any further details or comments.

---

> > > > > > ### Author Response · Authors · 2025-11-22
> > > > > > **Response to Reviewer p9ow (Second Round)**
> > > > > >
> > > > > > Thank you very much for your thoughtful assessment and for raising your score. We appreciate your recognition that our ablation study is now much more comprehensive and that the new experiments demonstrate the merit of our method.
> > > > > >
> > > > > >
> > > > > > > **with variable-length inputs, the models struggle to converge because of high variance—this remains a substantial limitation of the approach.**
> > > > > >
> > > > > > The convergence instability arises because Stages 2 and 3 rely on weak proxy signals, where the compounded variance of noisy supervision and fluctuating input sizes destabilizes the joint optimization of ranking and budgeting. We believe scaling up FrameOracle-41K will overcome this, as strong, explicit ground-truth supervision anchors the learning process, enabling the model to generalize across variable-length inputs without being disrupted by the high variance inherent in proxy-based training.
> > > > > >
> > > > > > > **Regarding the comparison with MovieChat and Rewind: if you present the frames, it is important to also show the number of tokens consumed.**
> > > > > >
> > > > > > We provide a step-by-step process to compute the token number for our reproduced LLaVA-OneVision+MovieChat and LLaVA-OneVision+FrameOracle.
> > > > > >
> > > > > > * LLaVA-OneVision+MovieChat: `short_memory_length = 18, long_memory_length = 64, sliding_window_length = 8, merge_frame_length = 2, mm_spatial_pool_stride = 2, Encoder output = 729`.
> > > > > >
> > > > > > Each frame produces $N = 27 \times 27 = 729$ tokens (LLaVA-OneVision).
> > > > > >
> > > > > > With `mm_spatial_pool_stride=2`, the grid is downsampled using ceiling logic (padding):
> > > > > >
> > > > > > $$
> > > > > > \text{Grid Size} = \lceil 27 / 2 \rceil \times \lceil 27 / 2 \rceil = 14 \times 14 = 196 \text{ Tokens/Frame} \\
> > > > > > $$
> > > > > >
> > > > > > Since long-term buffer $R^{L}=64$, long-term memory has
> > > > > >
> > > > > > $$
> > > > > > T_{\text{long}} = R^{L} \times \text{Grid Size} = \mathbf{12,544 \text{ Tokens}}
> > > > > > $$
> > > > > >
> > > > > > In global task of MovieChat-1K, it only uses long-term tokens, so $T_{\text{MovieChat}} = T_{\text{long}} = \mathbf{12,544}$
> > > > > >
> > > > > > * LLaVA-OneVision+FrameOracle: `Encoder output = 729, num_frames=15.6, mm_spatial_pool_stride = 2`
> > > > > >
> > > > > > With the same `mm_spatial_pool_stride=2`, the grid size is the same as $\text{Grid Size} = 196 \text{ Tokens/Frame}$.
> > > > > >
> > > > > > $$
> > > > > > T_{\text{FrameOracle}} = \text{Grid Size} \times n_{\text{frames}}= 196 \times 15.6 = \mathbf{3{,}057.6}
> > > > > > $$
> > > > > >
> > > > > > Overall, $T_{\text{MovieChat}} = \mathbf{12,544} ,   T_{\text{FrameOracle}} = \mathbf{3{,}057.6}$.
> > > > > >
> > > > > > ---
> > > > > > In another experiment, we applied FrameOracle to Qwen3-VL-8B, and the results are shown in the table below, demonstrating that FrameOracle improves performance even on the newest state-of-the-art VLM.
> > > > > >
> > > > > > | Model             | Frames     | NExTQA_OE_val | NExTQA_OE_test | NExTQA_MC   | Perception | LVB  | Video-MME | EgoSchema | MLVU | Avg  |
> > > > > > |-------------------|------------|--------|----------|------|-------------|------|------------|------------|------|------|
> > > > > > | **Qwen3-VL-8B**        | 32         | 26.0   | 31.1     | 76.6 | 67.5        | 63.3 | 66.9       | 70.8       | 63.6 | 58.2 |
> > > > > > | **+ FrameOracle** | 32→20.9    | 26.6   | 32.3     | 76.1 | 68.2        | 64.0 | 67.3       | 71.4       | 62.9 | 58.6 |
> > > > > > | **+ FrameOracle** | 128→27.8   | **28.1**   | **33.8**     | **77.3** | **69.0**        | **65.2** | **69.1**       | **72.3**       | **66.3** | **60.1** |
> > > > > >
> > > > > > We are currently running two additional ablation studies: (1) Jointly training Stage 2 and Stage 3. (2) Replacing the visual backbone with VideoLLaMA3. We will report the results in this thread as soon as they are available.

---

> > > > > > > ### Author Response · Authors · 2025-11-26
> > > > > > > **Response to Reviewer p9ow (Second Round - 2)**
> > > > > > >
> > > > > > > Dear Reviewer p9ow,
> > > > > > >
> > > > > > > We conducted an additional ablation study showing that joint training of the stages degrades performance, while our proposed staged curriculum preserves stability and improves downstream results.
> > > > > > >
> > > > > > > Specifically, when Stage 2 (Rank Head) and Stage 3 (K Head) are trained jointly, the **K Head collapses**, predicting near-maximum frames (i.e., failing to perform meaningful frame reduction), and the **Rank Head becomes unstable**, with Kendall-τ fluctuating between −0.4 and +0.6. This instability arises because the two objectives interfere during joint optimization: immature Rank predictions generate noisy top-K subsets that destabilize K-Head learning, and the resulting unstable K outputs further corrupt Rank learning, creating a feedback loop that prevents either head from converging.
> > > > > > >
> > > > > > > To isolate the impact on ranking, we evaluated the Qwen2.5-VL-3B backbone with 32-frame inputs. First, Kendall-τ correlation with ground-truth frame importance computed via leave-one-out (LOO) VLM loss **dropped from 0.5367 (Stage 2 alone) to 0.2313 (joint Stage 2 + 3)**, confirming that joint training substantially weakens ranking. Second, fixed 16-frame selection showed lower accuracy for the jointly trained model on all standard benchmarks, consistent with its degraded ranking (see Table below).
> > > > > > >
> > > > > > > These results confirm that staged training is essential for stable Rank Head learning and better downstream performance.
> > > > > > >
> > > > > > > | Setting | Frame | NExTQA_OE_val | NExTQA_OE_test | NExTQA_MC | Perception | LongVideoBench | VideoMME | EgoSchema | MLVU |
> > > > > > > | :--- | :---: | :---: | :---: | :---: | :---: | :---: | :---: | :---: | :---: |
> > > > > > > | Stage 2+3 (joint) | 32→16 | 24.0 | 27.8 | 72.2 | 63.8 | 50.1 | 54.3 | 49.8 | 52.2 |
> > > > > > > | Stage 2 alone | 32→16  | **24.8** | **29.5** | **73.0** | **64.7** | **51.9** | **55.7** | **52.2** | **54.8** |

---

> > > > > > > > ### Comment · Reviewer_p9ow · 2025-11-26
> > > > > > > > **Final decision**
> > > > > > > >
> > > > > > > > I thank the reviewer for the extra ablations. However, I think these new ablations confirm what was already demonstrated in the previous extension. I'll keep my score as it is; I've already increased it once. The method is interesting, but I believe if scaling is a bottleneck, its utility is limited. In addition, as mentioned in previous comments, a tradeoff must be made between temporal fidelity and more efficient video semantic understanding, which is impractical for many real applications, given the generic use of VLMs. From these concerns, in addition to the system's complexity with all the training stages and additional modules, I will keep my score as it is, and wish the authors good luck with the remaining reviews.

---

> > > > > > > > > ### Author Response · Authors · 2025-12-03
> > > > > > > > > **Response to Reviewer p9ow (Third Round)**
> > > > > > > > >
> > > > > > > > > Thank you again for the thoughtful feedback and for raising the score.
> > > > > > > > >
> > > > > > > > > As you requested in Question 7, we ran an ablation study replacing the Stage-2/3 backbone (Qwen2.5-VL-3B) with the stronger VideoLLaMA3-7B to assess how the selector behaves with a different visual backbone. For fairness, both variants were evaluated using the same downstream pipeline (LLaVA-OneVision-7B for inference) on the same benchmarks. As shown in the table below, replacing Qwen2.5-VL-3B with VideoLLaMA3-7B has minimal effect on the predicted frame counts since both selectors produce nearly identical values (13.9 versus 14.2 on average). Both variants also outperform the full-frame LLaVA-OneVision baseline, demonstrating that FrameOracle generalizes well across different visual backbones. In addition, the selector trained with VideoLLaMA3-7B achieves stronger performance than the one trained with Qwen2.5-VL-3B across benchmarks. This improvement is due to VideoLLaMA3-7B’s stronger visual representations, which provide richer information about events spanning longer periods of time. As a result, this selector is better at choosing frames that capture high-level or long-range context, leading to larger gains on long-video benchmarks.
> > > > > > > > >
> > > > > > > > >
> > > > > > > > > | Model | Frame | NExTQA_OE_val | NExTQA_OE_test | NExTQA_MC | Perception | LongVideoBench | VideoMME | EgoSchema | MLVU |
> > > > > > > > > | :--- | :---: | :---: | :---: | :---: | :---: | :---: | :---: | :---: | :---: |
> > > > > > > > > | LLaVA-OneVision (Baseline) | 16 | 14.6 | 16.7 | 78.2 | 56.4 | 55.0 | 56.1 | 60.8 | 60.9 |
> > > > > > > > > | +FrameOracle (Qwen2.5-VL-3B) | 64→13.9 | 16.5 | **19.0** | **78.5** | 56.9 | 56.5 | 58.1 | 63.4 | 63.7 |
> > > > > > > > > | +FrameOracle (VideoLLaMA3-7B) | 64→14.2  | **16.7** | 18.9 | **78.5** | **57.0** | **57.4** | **58.4** | **64.0** | **65.1** |

---

> > > ### Comment · Reviewer_p9ow · 2025-11-21
> > >
> > > It is interesting to see how each stage alone causes performance loss but together they improve the model performance.

---

> > > > ### Comment · Reviewer_p9ow · 2025-11-21
> > > >
> > > > Also, it would be nice to have some experiments on why DinoV2 and not some other model? Why note some model which has visual and language understanding given the interaction with language queries? These will be useful to strengthen the work even further.

---

### Official Review · Reviewer_CCS3 · 2025-10-31

**Soundness:** 3
**Presentation:** 3
**Contribution:** 3
**Rating:** 6
**Confidence:** 5

**Summary:**

The paper proposes a lightweight and plug-and-play module capable of dynamically selecting a variable number of frames based on the difficulty of each question.

In addition, the authors introduce a curriculum-based training strategy to effectively train this frame selection module.

The paper also designs a data generation pipeline that provides the minimal set of frames required to answer each question, forming the first VideoQA dataset with such keyframe annotations.

**Strengths:**

1. The paper is detailed and clearly presented, with strong motivation and solid overall design.

2. A novel module is introduced to jointly predict the number of frames to select and frame-level importance scores, along with a carefully designed training paradigm. The method is validated across multiple backbones, showing good generalization.

3. The authors build a new data generation pipeline, providing the first VideoQA dataset with minimal keyframe annotations, which is a valuable contribution to the community.

**Weaknesses:**

1. The proposed multi-head design for predicting both frame count and frame importance is reasonable, but it depends on the backbone’s global reasoning capability. The paper should clarify whether different backbones lead to significantly different results.

2. It would be helpful to compare against a strategy that fixes or predicts a total information budget (instead of a frame count) as the selection target — would such a formulation be more reasonable?

3. The paper adopts a curriculum learning scheme with four progressive training stages, and the ablation study supports each stage’s usefulness. However, is such staged training truly necessary? Could joint training achieve similar results? A comparison would make the claim more convincing.

4. Since the selected frames are all highly important, do they sometimes concentrate around similar patterns, causing redundancy in visual information? A discussion or visualization could help clarify this.

**Questions:**

Please see the weakness.

---

> ### Author Response · Authors · 2025-11-21
> **Response to Reviewer CCS3**
>
> We are grateful for your thoughtful review and for acknowledging the paper’s clear presentation, robust method design, and the novelty and value of our contribution to the community.
>
> > **The paper should clarify whether different backbones lead to significantly different results.**
>
> We are running an ablation using VideoLLaMA3 as the backbone in Stages 2 and 3. The experiments are underway, and we will share the results in this thread as soon as they are ready, before the official discussion deadline. We will also include them in the next revision of the paper.
>
> > **It would be helpful to compare against a strategy that fixes or predicts a total information budget (instead of a frame count) as the selection target ...**
>
> B-VLLM [1] is the only fixed-information-budget method we found that is relevant for comparison. It samples videos at 1 fps and applies a fixed 512-token visual budget to select frames and spatial tokens. B-VLLM can operate as a standalone VLLM or integrate with existing VLLMs; its strongest performance is reported in the standalone setting using Qwen2 as the backbone LLM. For fairness, we compared this configuration against LLaVA-Video+FrameOracle, which also uses Qwen2, and further evaluate both under a unified backbone (VideoLLaMA2) to isolate the effect of the selection mechanism.
>
> | Model  | Perception | VideoMME | EgoSchema |
> | :--- | :---: | :---: | :---: |
> | VideoLLaMA2+B-VLLM | 48.0  | 44.4 | 44.3 |
> | VideoLLaMA2+FrameOracle | **53.4** | **54.1** | **51.8** |
> | B-VLLM | 52.1  | 53.5 | 51.9 |
> | LLaVA-Video+FrameOracle | **65.1** | **61.6** | **55.2** |
>
> In both settings, FrameOracle consistently outperforms B-VLLM, showing that learning a per-instance frame-count budget is an effective way for adaptive information control in video LLMs, without token-level merging or architectural changes.
>
> To our knowledge, no prior work explicitly *predicts* per-instance information budgets for video understanding. This is likely because (1) adaptive budget prediction is still underexplored, even for simpler units like frame count, and (2) “information” is hard to define and operationalise in a model-agnostic way. If there are relevant works we missed, we would greatly appreciate pointers and will include comparisons in the rebuttal.
>
> > **Is such staged training truly necessary? Could joint training achieve similar results?**
>
> Our four-stage training is primarily an optimisation strategy, not an architectural requirement. Stages 2 and 3 both use the downstream VLM loss as soft supervision for different purposes: Stage 2 trains the Rank Head via leave-one-out (LOO) loss differences to identify important frames, while Stage 3 uses the same loss over different top-K subsets (from Stage 2) to train the K Head to predict the number of frames needed for each video–query pair.
>
> Training both heads jointly from scratch under this LOO supervision is unstable: inaccurate rankings make top-K losses noisy, and poor K predictions degrade the ranking signal, preventing either head from converging. To test this, we are running an ablation jointly optimising Stages 2 and 3 and will report results before the discussion deadline **(UPDATE: See the thread below for the new ablation)**.
>
> Stage 4 already fine-tunes both heads jointly using strong supervision from FrameOracle-41K, once the soft-supervised stages have provided a stable initialisation. Our design does not forbid joint training; it simply delays it until supervision is reliable.
>
> > **Since the selected frames are all highly important, do they sometimes concentrate around similar patterns, causing redundancy in visual information?**
>
> Our selector is not pointwise. All frames interact through a Transformer encoder with self-attention before scoring. This allows the model to perceive global context—if multiple frames convey similar semantic information, the attention mechanism aggregates their features, enabling the Rank Head to score them conditionally (e.g., highlighting the most representative frame while suppressing near-duplicates) rather than treating them independently.
>
> In stage 3, the K Head is explicitly trained to balance accuracy and frame count (Equation 2). If the Rank Head were to assign high scores to a cluster of redundant frames, adding more of them would yield diminishing returns on the VLM task loss. Consequently, the K Head learns to predict a cut-off point (K*) that halts selection once the information saturates, effectively filtering out the "tail" of redundant high-scoring frames.
>
> In Stage 4, the model is fine-tuned on FrameOracle-41K, where the ground truth is explicitly defined as the "minimal sufficient set" of keyframes. This provides a direct, strong supervision signal that teaches the model to penalise redundancy and select only the necessary evidence required to answer the query.
>
> ---
>
> [1] B-VLLM: A Vision Large Language Model with Balanced Spatio-Temporal Tokens. ICCV'25.

---

> ### Author Response · Authors · 2025-11-26
> **Response to Reviewer CCS3 (2)**
>
> Dear Reviewer CCS3,
>
> > **Is such staged training truly necessary? Could joint training achieve similar results?**
>
> For this question, we conducted an additional ablation study showing that joint training of the stages degrades performance, while our proposed staged curriculum preserves stability and improves downstream results.
>
> Specifically, when Stage 2 (Rank Head) and Stage 3 (K Head) are trained jointly, the **K Head collapses**, predicting near-maximum frames (i.e., failing to perform meaningful frame reduction), and the **Rank Head becomes unstable**, with Kendall-τ fluctuating between −0.4 and +0.6. This instability arises because the two objectives interfere during joint optimization: immature Rank predictions generate noisy top-K subsets that destabilize K-Head learning, and the resulting unstable K outputs further corrupt Rank learning, creating a feedback loop that prevents either head from converging.
>
> To isolate the impact on ranking, we evaluated the Qwen2.5-VL-3B backbone with 32-frame inputs. First, Kendall-τ correlation with ground-truth frame importance computed via leave-one-out (LOO) VLM loss **dropped from 0.5367 (Stage 2 alone) to 0.2313 (joint Stage 2 + 3)**, confirming that joint training substantially weakens ranking. Second, fixed 16-frame selection showed lower accuracy for the jointly trained model on all standard benchmarks, consistent with its degraded ranking (see Table below).
>
> These results confirm that staged training is essential for stable Rank Head learning and better downstream performance.
>
> | Setting | Frame | NExTQA_OE_val | NExTQA_OE_test | NExTQA_MC | Perception | LongVideoBench | VideoMME | EgoSchema | MLVU |
> | :--- | :---: | :---: | :---: | :---: | :---: | :---: | :---: | :---: | :---: |
> | Stage 2+3 (joint) | 32→16 | 24.0 | 27.8 | 72.2 | 63.8 | 50.1 | 54.3 | 49.8 | 52.2 |
> | Stage 2 alone | 32→16  | **24.8** | **29.5** | **73.0** | **64.7** | **51.9** | **55.7** | **52.2** | **54.8** |

---

> > ### Comment · Area_Chair_PFRj · 2025-11-26
> >
> > Dear reviewer CCS3,
> >
> > Could you please take a look at the author's response to your comments and leave your feedback?
> >
> > AC

---

> ### Author Response · Authors · 2025-12-03
> **Response to Reviewer CCS3 (3)**
>
> > **The paper should clarify whether different backbones lead to significantly different results.**
>
> As you requested, we ran an ablation study replacing the Stage-2/3 backbone (Qwen2.5-VL-3B) with the stronger VideoLLaMA3-7B to assess how the selector behaves with a different visual backbone. For fairness, both variants were evaluated using the same downstream pipeline (LLaVA-OneVision-7B for inference) on the same benchmarks. As shown in the table below, replacing Qwen2.5-VL-3B with VideoLLaMA3-7B has minimal effect on the predicted frame counts since both selectors produce nearly identical values (13.9 versus 14.2 on average). Both variants also outperform the full-frame LLaVA-OneVision baseline, demonstrating that FrameOracle generalizes well across different visual backbones. In addition, the selector trained with VideoLLaMA3-7B achieves stronger performance than the one trained with Qwen2.5-VL-3B across benchmarks. This improvement is due to VideoLLaMA3-7B’s stronger visual representations, which provide richer information about events spanning longer periods of time. As a result, this selector is better at choosing frames that capture high-level or long-range context, leading to larger gains on long-video benchmarks.
>
>
> | Model | Frame | NExTQA_OE_val | NExTQA_OE_test | NExTQA_MC | Perception | LongVideoBench | VideoMME | EgoSchema | MLVU |
> | :--- | :---: | :---: | :---: | :---: | :---: | :---: | :---: | :---: | :---: |
> | LLaVA-OneVision (Baseline) | 16 | 14.6 | 16.7 | 78.2 | 56.4 | 55.0 | 56.1 | 60.8 | 60.9 |
> | +FrameOracle (Qwen2.5-VL-3B) | 64→13.9 | 16.5 | **19.0** | **78.5** | 56.9 | 56.5 | 58.1 | 63.4 | 63.7 |
> | +FrameOracle (VideoLLaMA3-7B) | 64→14.2  | **16.7** | 18.9 | **78.5** | **57.0** | **57.4** | **58.4** | **64.0** | **65.1** |

---

### Official Review · Reviewer_otKT · 2025-11-02

**Soundness:** 2
**Presentation:** 3
**Contribution:** 2
**Rating:** 4
**Confidence:** 3

**Summary:**

The paper introduces FrameOracle, a lightweight and plug-and-play frame selection module that dynamically determines both the most relevant frames and the number of frames required for a given video understanding task. To support its training, the authors also present FrameOracle-41K, the first large-scale VideoQA dataset annotated with keyframes that specify the minimal frame subset needed to answer each question. The proposed approach is evaluated across six benchmarks and compared against five vision-language models (VLMs), demonstrating strong efficiency gains while preserving task accuracy.

**Strengths:**

- The paper introduces FrameOracle-41K, a large-scale dataset specifically created for keyframe selection in VideoQA, with annotations indicating the minimal set of frames required to answer each question.

- The proposed method improves computational efficiency by reducing the number of processed frames, while still maintaining comparable or better accuracy than full-frame baselines.

- FrameOracle outperforms existing keyframe selection methods, showing better frame relevance and stronger downstream task performance across multiple benchmarks.

**Weaknesses:**

- The data generation process heavily relies on another agent model for producing keyframe annotations, which raises concerns about potential bias, annotation noise, and the dependency of the dataset quality on the agent’s capabilities.

- The data generation pipeline appears relatively simple and lacks clear novelty. How does it differ from existing data generation approaches, and what unique contributions does it offer?

- The training process consists of four distinct stages, which adds considerable complexity to the pipeline and may hinder scalability and ease of adoption in practical settings.

**Questions:**

How does a simple baseline, such as uniform sampling, perform in comparison? For instance, when FrameOracle reduces the number of frames from 16 to 10.4 on average, how does uniform sampling of 10 frames from the original 16-frame sequence compare in terms of performance?

---

> ### Author Response · Authors · 2025-11-21
> **Response to Reviewer otKT (1)**
>
> Thank you for your thoughtful review and for recognizing the novelty of FrameOracle-41K, the substantial compute savings without loss of accuracy, and FrameOracle’s superior performance over prior keyframe selection methods.
>
> > **The data generation process heavily relies on another agent ... which raises concerns about potential bias ...**
>
> To address your concern, we conducted a **human verification study** on 4,000 randomly sampled instances (~10% of the dataset). Ten annotators participated, each reviewing 800 samples, with two annotators per sample. A sample was considered *verified* only if both annotators correctly answered the question using the provided keyframes. The detailed result is shown in the table below. Inter-annotator agreement reaches 94%, indicating significant consistency. The overall human-verified accuracy is **93.3%** (3,732/4,000), confirming that most keyframe sets provide sufficient evidence for human reasoning.
>
> | Agreement Type     | Count | Percentage |
> |--------------------|-------|-------------|
> | Both correct       | 3,732 | 93.3% |
> | One correct / one wrong | 240 | 6.0% |
> | Both wrong         | 28 | 0.7% |
>
> For all verified samples, annotators further labeled each keyframe set as:
> * Excessive: ≥5 frames beyond minimal sufficiency,
> * Just-right: within ±4 frames of minimal sufficiency,
> * Insufficient: missing critical evidence.
>
> The breakdown is: **97.2%** Just-right (3,628/3,732), **2.6%** Excessive (98/3,732), and **0.2%** Insufficient (6/3,732). These findings show that **the vast majority of annotations are both semantically accurate and frame-efficient**, demonstrating that FrameOracle-41K provides high-quality supervision.
>
> Moreover, we highlight that FrameOracle-41K was created using a **rigorous two-stage verification** pipeline to mitigate model bias. In Stage II, we required **three-model consensus** across Qwen2.5-VL-72B, LLaVA-OneVision-72B, and LLaVA-Video-72B, three large VLMs with **distinct architectures, visual encoders, and language backbones**. A sample was retained only if all three models independently produced the correct answer using just the provided keyframes, ensuring that retained samples reflect consistent, cross-model reasoning rather than any single model’s bias.
>
> > **The data generation pipeline appears relatively simple ...**
>
> We emphasize that the true value of a dataset lies in its **contribution to the research community** rather than the complexity of its generation pipeline. FrameOracle-41K makes a fundamentally new contribution as, to our knowledge, the **first dataset providing frame-level minimal evidence annotations** for VideoQA.
>
> Its novelty lies in both the annotation design and the data generation process.
>
> * **Stage I: Iterative frame-level evidence localization:** Unlike existing pipelines that annotate at the clip or answer level (e.g., VideoInstruct100K [1], VCG+112K [2]), our agent iteratively refines temporal segments to identify the minimal subset of frames required to answer each question—introducing a new paradigm for fine-grained, interpretable visual reasoning.
> * **Stage II: Multi-model verification for robustness:** To mitigate bias from any single model, we verify each keyframe set using three VLMs with different architectures. Only sets that produce correct answers across all three models are retained, greatly improving the reliability of the annotations compared to single-model pipelines [3].
> * **New VideoQA paradigms:** FrameOracle-41K enables new training and evaluation paradigms, such as evidence-based learning and interpretable QA, which are not possible with existing datasets.
>
> > **The training process consists of four distinct stages, which adds considerable complexity ...**
>
> The four-stage training design addresses the **lack of frame-level supervision** in existing datasets. To our knowledge, FrameOracle-41K is the first dataset providing such ground truth, but with only 41K instances, much smaller than typical video-language corpora, training a selector from scratch would risk poor convergence and overfitting. Consequently, the selector first learns under **soft or proxy supervision**. Stage 1 uses SigLIP-based text–visual alignment to initialize the cross-modal encoder. Stages 2–3 apply weak multimodal-LLM objectives for ranking and frame-count prediction without ground-truth keyframes. Jointly optimizing these objectives from scratch leads to unstable convergence and redundant frame selection. To verify this, we are running an ablation where Stage 2 and 3 are trained jointly. We will provide the results in this thread as soon as they are ready **(UPDATE: See the thread below for the new ablation).**
>
> Moreover, the four-stage training pipeline is modular and self-contained, making it straightforward to reproduce or adapt. At deployment, only the lightweight **80M-parameter** selector is required, and it can be plugged into any VLM as a preprocessing module without further fine-tuning.

---

> > ### Author Response · Authors · 2025-11-21
> > **Response to Reviewer otKT (2)**
> >
> > > **When FrameOracle reduces the number of frames from 16 to 10.4 on average, how does uniform sampling of 10 frames from the original 16-frame sequence compare in terms of performance?**
> >
> > To address your question, we conducted an ablation on LLaVA-OneVision-7B, comparing FrameOracle with uniform sampling under the same frame budget. We uniformly sampled 10 frames from the original 16-frame inputs, matching FrameOracle’s average of 10.4 frames, and evaluated both methods on six benchmarks. As shown below, FrameOracle consistently outperforms uniform sampling, confirming that the gains come not from reducing the number of frames, but from selecting semantically relevant, query-conditioned evidence.
> >
> > | Model | Frame | NExTQA_OE_val | NExTQA_OE_test | NExTQA_MC | Perception | LongVideoBench | VideoMME | EgoSchema | MLVU |
> > | :--- | :---: | :---: | :---: | :---: | :---: | :---: | :---: | :---: | :---: |
> > | LLaVA-OneVision-7B | 16→10  | 8.8 | 11.5 | 73.5 | 53.8 | 53.2 | 54.1 | 60.0 | 55.8 |
> > | +FrameOracle | 16→10.4 | **16.1** | **17.8** | **77.6** | **56.5** | **55.5** | **56.0** | **62.4** | **60.2** |
> >
> > ---
> >
> > [1] Video-ChatGPT: Towards Detailed Video Understanding via Large Vision and Language Models. ACL'24
> >
> > [2] Video-ChatGPT+: Integrating Image and Video Encoders for Enhanced Video Understanding. arXiv'24
> >
> > [3] ShareGPT4Video: Improving Video Understanding and Generation with Better Captions. NeurIPS'24

---

> ### Author Response · Authors · 2025-11-26
> **Response to Reviewer otKT (3)**
>
> Dear Reviewer otKT,
>
> > **The training process consists of four distinct stages, which adds considerable complexity ...**
>
> For this question, we conducted an additional ablation study showing that joint training of the stages degrades performance, while our proposed staged curriculum preserves stability and improves downstream results.
>
> Specifically, when Stage 2 (Rank Head) and Stage 3 (K Head) are trained jointly, the **K Head collapses**, predicting near-maximum frames (i.e., failing to perform meaningful frame reduction), and the **Rank Head becomes unstable**, with Kendall-τ fluctuating between −0.4 and +0.6. This instability arises because the two objectives interfere during joint optimization: immature Rank predictions generate noisy top-K subsets that destabilize K-Head learning, and the resulting unstable K outputs further corrupt Rank learning, creating a feedback loop that prevents either head from converging.
>
> To isolate the impact on ranking, we evaluated the Qwen2.5-VL-3B backbone with 32-frame inputs. First, Kendall-τ correlation with ground-truth frame importance computed via leave-one-out (LOO) VLM loss **dropped from 0.5367 (Stage 2 alone) to 0.2313 (joint Stage 2 + 3)**, confirming that joint training substantially weakens ranking. Second, fixed 16-frame selection showed lower accuracy for the jointly trained model on all standard benchmarks, consistent with its degraded ranking (see Table below).
>
> These results confirm that staged training is essential for stable Rank Head learning and better downstream performance.
>
> | Setting | Frame | NExTQA_OE_val | NExTQA_OE_test | NExTQA_MC | Perception | LongVideoBench | VideoMME | EgoSchema | MLVU |
> | :--- | :---: | :---: | :---: | :---: | :---: | :---: | :---: | :---: | :---: |
> | Stage 2+3 (joint) | 32→16 | 24.0 | 27.8 | 72.2 | 63.8 | 50.1 | 54.3 | 49.8 | 52.2 |
> | Stage 2 alone | 32→16  | **24.8** | **29.5** | **73.0** | **64.7** | **51.9** | **55.7** | **52.2** | **54.8** |

---

> > ### Comment · Area_Chair_PFRj · 2025-11-26
> >
> > Dear reviewer otKT,
> >
> > Could you please take a look at the author's response to your comments and leave your feedback?
> >
> > AC

---

### Official Review · Reviewer_m9jf · 2025-11-03

**Soundness:** 3
**Presentation:** 2
**Contribution:** 3
**Rating:** 6
**Confidence:** 2

**Summary:**

Authors propose FrameOracle, a lightweight, plug-and-play selector for video-VLMs that predicts both which frames are relevant to a given query and how many frames are actually needed, tackling the inefficiency of uniform or fixed-budget sampling. It’s trained via a four-stage curriculum that begins with weak proxy signals and culminates in supervised fine-tuning on a new dataset, FrameOracle-41K, which supplies keyframe annotations specifying the minimal sufficient frames per question. Across five VLMs and six benchmarks, FrameOracle cuts 16-frame inputs to ~10.4 with no accuracy loss and trims 64 candidates to ~13.9 while improving accuracy by ~1.4%.

**Strengths:**

- Performance Gains

FrameOracle reduces frame usage while maintaining or improving accuracy across six diverse benchmarks and five different video-language models


- Plug-and-Play Generalization without Co-Training

Unlike most keyframe selection methods, FrameOracle operates independently of the base VLM, requiring no co-training or model-specific tuning — showing strong transferability and making it highly practical for real-world deployment

- A Novel Dataset (FrameOracle-41K)

The paper contributes a large, purpose-built dataset with keyframe supervision, enabling both training for adaptive frame selection

**Weaknesses:**

- Marginal Gains at Larger Compute Budgets

When starting from a large candidate pool (e.g., 64 frames), efficiency gains diminish, achieving only modest FLOP and latency reductions, which limits its benefit for already optimized pipelines

- Performance on Fine-Grained Temporal Tasks

FrameOracle underperforms heuristic methods like KFC on datasets such as MLVU, which require precise temporal reasoning and multi-event grounding.

**Questions:**

Can you show more examples from the dataset itself.

---

> ### Author Response · Authors · 2025-11-21
> **Response to Reviewer m9jf**
>
> We are grateful for your encouraging review and for highlighting our strong performance, FrameOracle’s plug-and-play generalization, and the novelty and value of the FrameOracle-41K dataset.
>
> > **Marginal Gains at Larger Compute Budgets**
>
> To clarify this point, the table below breaks down the FLOPs for the 16 → 10.4 and 64 → 13.9 settings. The total compute increases from **110.98 to 167.67** TFLOPs (+56.7), but **~90%** of this increase (+50.98 TFLOPs) comes from the **backbone VLM**, not from FrameOracle. The selector (DINOv2 + FrameOracle) accounts for only **~10%** (+5.71 TFLOPs). Because VLM inference dominates the compute budget and scales nearly linearly with the number of retained frames, most of the added cost is inherent to the backbone, not the selector.
>
> Importantly, the 64 → 13.9 configuration, despite starting from a much larger frame pool, still retains fewer frames and achieves **~10% lower total compute and higher accuracy** compared to the 16-frame baseline.
>
> | Setting | DINOv2 | FrameOracle | VLM | Total |
> | :--- | :---: | :---: | :---: | :---: |
> | 16 → 10.4 | 1.87 | 0.00026 | 109.11 | 110.98 |
> | 64 → 13.9 | 7.58 | 0.001   | 160.09 | 167.67 |
>
> > **Performance on Fine-Grained Temporal Tasks**
>
> The performance gap on MLVU (Section 5.2, RQ2) is expected. KFC’s hand-tuned 'diversity' term explicitly selects visually distinct frames, which works well for narrow tasks like counting or ordering scattered events, dominant in MLVU. FrameOracle, in contrast, learns frame sufficiency directly from data, predicting both relevance and the number of frames needed. This learned approach generalizes across diverse video types and VideoQA tasks, adapting to varying information densities, as demonstrated on several other benchmarks.
>
> > **Can you show more examples from the dataset itself.**
>
> Thank you for the suggestion. We added additional examples from our FrameOracle-41K dataset in Appendix H of the revised paper.

---

> > ### Comment · Area_Chair_PFRj · 2025-11-26
> >
> > Dear reviewer m9jf,
> >
> > Could you please take a look at the author's response to your comments and leave your feedback?
> >
> > AC

---

### Author Response · Authors · 2025-12-03
**General Response to Reviewers (2/2)**

**3.** `[CCS3, p9ow]`: **Influence of Visual Backbone Choices:** Do different visual backbones choices significantly affect FrameOracle’s behavior?

We ran an ablation study replacing the Stage-2/3 backbone (Qwen2.5-VL-3B) with the stronger VideoLLaMA3-7B to assess how the selector behaves with a different visual backbone. For fairness, both variants were evaluated using the same downstream pipeline (LLaVA-OneVision-7B for inference) on the same benchmarks. As shown in the table below, replacing Qwen2.5-VL-3B with VideoLLaMA3-7B has minimal effect on the predicted frame counts since both selectors produce nearly identical values (13.9 versus 14.2 on average). Both variants also outperform the full-frame LLaVA-OneVision baseline, demonstrating that FrameOracle generalizes well across different visual backbones. In addition, the selector trained with VideoLLaMA3-7B achieves stronger performance than the one trained with Qwen2.5-VL-3B across benchmarks. This improvement is due to VideoLLaMA3-7B’s stronger visual representations, which provide richer information about events spanning longer periods of time. As a result, this selector is better at choosing frames that capture high-level or long-range context, leading to larger gains on long-video benchmarks.

| Model | Frame | NExTQA_OE_val | NExTQA_OE_test | NExTQA_MC | Perception | LongVideoBench | VideoMME | EgoSchema | MLVU |
| :--- | :---: | :---: | :---: | :---: | :---: | :---: | :---: | :---: | :---: |
| LLaVA-OneVision (Baseline) | 16 | 14.6 | 16.7 | 78.2 | 56.4 | 55.0 | 56.1 | 60.8 | 60.9 |
| +FrameOracle (Qwen2.5-VL-3B) | 64→13.9 | 16.5 | **19.0** | **78.5** | 56.9 | 56.5 | 58.1 | 63.4 | 63.7 |
| +FrameOracle (VideoLLaMA3-7B) | 64→14.2  | **16.7** | 18.9 | **78.5** | **57.0** | **57.4** | **58.4** | **64.0** | **65.1** |

---

**4.** `p9ow` **Necessity of Intermediate Stages:** Can the selector achieve comparable performance if trained only with Stage 1 and Stage 4?

We conducted an ablation on Stage 1+4 only, under (1) fixed-frame and (2) adaptive frame settings. Stage 1+4 consistently underperformed Stage 1+2 and failed in the adaptive-K setting, with the K Head collapsing and overfitting dataset statistics. These results support the curriculum design: Stages 2 and 3 are essential for stable ranking and frame-count prediction, while Stage 4 provides supervised refinement.

---

**5.** `CCS3` **Comparison with Token-Level Information Budgeting:** How does FrameOracle compare with the methods that fix or predict a total information budget (e.g., token-level selection)?

We compared FrameOracle with B-VLLM, a representative fixed-token-budget method that selects spatial–temporal tokens under a 512-token constraint. Under both a unified backbone (VideoLLaMA2) and the configuration where B-VLLM performs best, FrameOracle consistently achieves higher accuracy while requiring no token merging or architectural modifications. These results show that dynamically predicting a per-instance frame-count budget is a more effective and simpler strategy for controlling visual information than enforcing a fixed token budget.

---

**6.** `p9ow` **Comparison with Long-Video Compression Methods:** How does FrameOracle compare with the memory-based long-video approaches?

We added experiments comparing FrameOracle with the long-video understanding/compression method, MovieChat. Using a unified backbone for fair evaluation, FrameOracle consistently outperforms this memory-based approach on MovieChat-1K and other long-video benchmarks, while using far fewer frames and tokens. These results demonstrate that adaptive keyframe selection is an efficient and competitive alternative to sequential compression for long-video reasoning.

---

**7.** `otKT ` **Comparison with Uniform Sampling:** How does FrameOracle compare to uniformly sampling 10 frames from the original 16-frame sequence?

We conducted an ablation on LLaVA-OneVision-7B, comparing FrameOracle with uniform sampling under the same frame budget. We uniformly sampled 10 frames from the original 16-frame inputs, matching FrameOracle’s average of 10.4 frames, and evaluated both methods on six benchmarks. The results show that FrameOracle consistently outperforms uniform sampling, confirming that the gains come not from reducing the number of frames, but from selecting semantically relevant, query-conditioned evidence.

---

### Author Response · Authors · 2025-12-03
**General Response to Reviewers (1/2)**

In this post, **we outline the reviewers' main questions and the experiments we conducted to address them.**

**1.** `[otKT, p9ow]` **Data Quality and Potential Bias:** Does the data-generation pipeline raise concerns about annotation bias or insufficient supervision quality?

To address this question, we conducted a human verification study on 4,000 randomly sampled instances (~10% of the dataset). Each sample was reviewed by two annotators and counted as verified only if both answered correctly using the provided keyframes. We observed **93.3%** human-verified accuracy and 94% inter-annotator agreement, indicating that the keyframes generally provide sufficient evidence for answering the questions.

| Agreement Type     | Count | Percentage |
|--------------------|-------|-------------|
| Both correct       | 3,732 | 93.3% |
| One correct / one wrong | 240 | 6.0% |
| Both wrong         | 28 | 0.7% |

For all verified samples, annotators further labeled each keyframe set as:
* Excessive: ≥5 frames beyond minimal sufficiency,
* Just-right: within ±4 frames of minimal sufficiency,
* Insufficient: missing critical evidence.

The breakdown is: **97.2%** Just-right (3,628/3,732), **2.6%** Excessive (98/3,732), and **0.2%** Insufficient (6/3,732). These findings show that **the vast majority of annotations are both semantically accurate and frame-efficient**, demonstrating that FrameOracle-41K provides high-quality supervision.

Moreover, we highlight that FrameOracle-41K was created using a **rigorous two-stage verification** pipeline to mitigate model bias. In Stage II, we required **three-model consensus** across three large VLMs with **distinct architectures, visual encoders, and language backbones**. A sample was retained only if all three models independently produced the correct answer using just the provided keyframes, ensuring that retained samples reflect consistent, cross-model reasoning rather than any single model’s bias.

---

**2.** `[otKT, CCS3, p9ow]` **Necessity of the Multi-Stage Curriculum:** Is the four-stage training pipeline truly required, or could a simpler joint training scheme achieve comparable stability and performance?

We conducted an ablation study to test whether the four-stage curriculum could be simplified via joint training. When Stage 2 (Rank Head) and Stage 3 (K Head) were trained together, training became unstable: the K Head collapsed, predicting near-maximum frame counts, and the Rank Head oscillated in Kendall-τ between −0.4 and 0.6. This instability arises because the two objectives interfere during joint optimization, generating noisy top-K signals that prevent either head from converging.

To isolate the impact on ranking, we evaluated the Qwen2.5-VL-3B backbone with 32-frame inputs. First, Kendall-τ correlation with ground-truth frame importance computed via leave-one-out (LOO) VLM loss **dropped from 0.5367 (Stage 2 alone) to 0.2313 (joint Stage 2 + 3)**, confirming that joint training substantially weakens ranking. Second, fixed 16-frame selection showed lower accuracy for the jointly trained model on all standard benchmarks, consistent with its degraded ranking (see Table below).

These results confirm that staged training is essential for stable Rank Head learning and better downstream performance.

| Setting | Frame | NExTQA_OE_val | NExTQA_OE_test | NExTQA_MC | Perception | LongVideoBench | VideoMME | EgoSchema | MLVU |
| :--- | :---: | :---: | :---: | :---: | :---: | :---: | :---: | :---: | :---: |
| Stage 2+3 (joint) | 32→16 | 24.0 | 27.8 | 72.2 | 63.8 | 50.1 | 54.3 | 49.8 | 52.2 |
| Stage 2 alone | 32→16  | **24.8** | **29.5** | **73.0** | **64.7** | **51.9** | **55.7** | **52.2** | **54.8** |

---

### Author Response · Authors · 2025-12-03
**Key Strengths and Paper Updates**

We sincerely thank all the reviewers for their valuable feedback. We are pleased to note that reviewer `p9ow` has increased their score following our clarifications.

In this post:

* We highlight the key strengths of our work as noted by the reviewers.
* We summarize the changes to the updated paper (marked in blue).

**(1) Key Strengths**

* Presentation and Motivation
    * `CCS3`: *"detailed and clearly presented, with strong motivation and solid overall design."*
* Method
    * `m9jf`: *"Plug-and-Play Generalization without Co-Training… FrameOracle operates independently of the base VLM… showing strong transferability and making it highly practical for real-world deployment."*
    * `CCS3`: *"A novel module is introduced to jointly predict the number of frames to select and frame-level importance scores, along with a carefully designed training paradigm."*
    * `p9ow`: *"Adaptive frame selection method… an extension of what current SOTA models are doing."*
* Dataset
    * `m9jf`: *"A Novel Dataset (FrameOracle-41K)… a large, purpose-built dataset with keyframe supervision, enabling training for adaptive frame selection."*
    * `otKT`: *"a large-scale dataset with annotations indicating the minimal set of frames required to answer each question."*
    * `CCS3` : *"The authors build a new data generation pipeline, providing the first VideoQA dataset with minimal keyframe annotations, which is a valuable contribution to the community."*
    * `p9ow`: *"the dataset is a significant contribution of this work."*
* Experiments & Results
    * `m9jf`: *"reduces frame usage while maintaining or improving accuracy across six diverse benchmarks and five different video-language models."*
    * `otKT`: *"improves computational efficiency by reducing the number of processed frames, while still maintaining comparable or better accuracy."*, *"outperforms existing keyframe selection methods, showing better frame relevance and stronger downstream task performance across multiple benchmarks."*
    * `p9ow`: *"considered many different datasets which increase the reliability of the method in the considered settings.", "The ablation study has become much more comprehensive", "the work is now significantly more complete", "the new experiments demonstrate that the method has merit"*

**(2) Changes to the Paper**
* Section 3: FrameOracle-41K Dataset
    * `otKT,p9ow`: Added a paragraph on the new human-verification study with a pointer to the Appendix for details (Section 3.1).
    * `p9ow`: Added Figure 3 showing the FrameOracle-41K question types and included the corresponding analysis (Section 3.2).

* Section 4: Method
    * `p9ow`: Clarified tokenizer usage during training and provided a clearer explanation of the cross-modal fusion mechanism (Section 4.1).

* Section 5: Experiments
    * `p9ow`: Added a new experiment applying FrameOracle to the new SOTA model Qwen3-VL-8B (Table 1).
    * `m9jf`: Clarified that when starting from a large candidate pool (e.g., 64 frames), the added compute comes almost entirely from the backbone VLM rather than the selector (Page 10, RQ3).

* Appendix
    * `p9ow`: Added additional dataset statistics and corresponding analysis (Appendix D).
    * `otKT,p9ow`: Added the new human-verification study and analysis validating FrameOracle-41K's data quality (Appendix E).
    * `otKT,p9ow`: Added two new ablation studies demonstrating the necessity of the proposed multi-stage training curriculum (Appendix F.1).
    * `CCS3,p9ow`: Added an ablation study demonstrating FrameOracle’s generalizability across different visual backbones (Appendix F.2).
    * `otKT`: Added an ablation study comparing FrameOracle to a uniform-sampling baseline with matched frame budgets, showing FrameOracle’s superior performance (Appendix F.3).
    * `p9ow`: Added an ablation study on how freezing the Rank Head in Stage 3 destabilizes ranking training and affects downstream behavior (Appendix F.4).
    * `CCS3`: Added a comparison of frame-level (FrameOracle) vs. token-level (B-VLLM) information budgeting, showing FrameOracle’s superiority (Appendix F.5).
    * `p9ow`: Added an experiment showing FrameOracle outperforms memory-based long-video understanding methods (Appendix F.6).
    * `m9jf`: Added additional visualization examples of FrameOracle-41K (Appendix H).

---

### Meta-Review · Area_Chair_2is2 · 2025-12-23

**Summary:**

As a strength, the paper contributes with a large VideoQA dataset (annotated with the minimal set of frames required to answer each question). This contribution has been acknowledged by all reviewers and is considered a significant contribution.  In addition to the dataset the authors proposed a method, called FrameOracle, which ranks available frames and estimates the minimal number of frames required. Its plug-and-play nature and combination with 5 video-language models is appreciated. However, the relevance of this contributed is disputed by the reviewers (see below).

The AC considers agrees with the critical reviewers and thinks that the technical contribution of FrameOracle is rather small. The scientific contribution is not clear, the method is rather complex (4 stages) and provides small gains over existing methods.  Doubts whether the method scales to larger videos remain. The dataset is considered a relevant contribution, however, it is insufficient to merit acceptation and the therefore the AC recommends rejection, and recommends sending the paper to the next major venue further incorporating the many recommendations of the reviewers.

**Reviewer Concerns:**

Most reviewer concerns have been addressed, even though some remain. However, none of the reviewers is very enthusiastic about the proposed method and its relevance /impact.

Addressed/remaining points:
m9jf. Marginal Gains at Larger Compute Budgets : addressed (though gains are very small in Table 2)
m9jf. Mediocre performance on fine-grained temporal tasks : not addressed (but authors correctly point out their superiority on the other 5 datasets)

otKT. The data generation process heavily relies on another agent. Addressed with human verification study
otKT. The data generation pipeline appears relatively simple and lacks clear novelty: AC agrees with authors that that is not major problem.
otKT. The training process consists of four distinct stages, which adds considerable complexity to the pipeline and may hinder scalability and ease of adoption in practical settings: this weak point remains.

CCS3. impact of backbone: addressed
CCS3.Better ablation 4 stages: addressed

P9ow has many critical points most of which are addressed by the authors. Although their points are addressed there are still the following remaining negative points: ‘While the improvements are meaningful, they are not groundbreaking’ and ‘The method is interesting, but I believe scaling is a bottleneck’. AC: the scaling remains not convincingly addressed.

**Reviewer Scores:**

The review scores at the end are:
m9jf:   6 ->6
otKT:   4 -> 4 (maybe 6)
CCS3: 6-> 6
p9ow:  2 -> 4

---

### Decision · Program_Chairs · 2026-01-26

Reject